# Interplay between Signaling Pathways and Tumor Microenvironment Components: A Paradoxical Role in Colorectal Cancer

**DOI:** 10.3390/ijms24065600

**Published:** 2023-03-15

**Authors:** Sonia Ben Hamouda, Khadija Essafi-Benkhadir

**Affiliations:** Laboratoire d’Epidémiologie Moléculaire et Pathologie Expérimentale, LR16IPT04, Institut Pasteur de Tunis, Université de Tunis El Manar, Tunis 1002, Tunisia

**Keywords:** colorectal cancer, tumor microenvironment, signaling pathways, effectors, dual function

## Abstract

The study of the tumor microenvironment (TME) has become an important part of colorectal cancer (CRC) research. Indeed, it is now accepted that the invasive character of a primary CRC is determined not only by the genotype of the tumor cells, but also by their interactions with the extracellular environment, which thereby orchestrates the development of the tumor. In fact, the TME cells are a double-edged sword as they play both pro- and anti-tumor roles. The interaction of the tumor-infiltrating cells (TIC) with the cancer cells induces the polarization of the TIC, exhibiting an antagonist phenotype. This polarization is controlled by a plethora of interconnected pro- and anti-oncogenic signaling pathways. The complexity of this interaction and the dual function of these different actors contribute to the failure of CRC control. Thus, a better understanding of such mechanisms is of great interest and provides new opportunities for the development of personalized and efficient therapies for CRC. In this review, we summarize the signaling pathways linked to CRC and their implication in the development or inhibition of the tumor initiation and progression. In the second part, we enlist the major components of the TME and discuss the complexity of their cells functions.

## 1. Introduction

Colorectal cancer (CRC) is one of the most common malignancies worldwide [1]. According to GLOBOCAN data, colorectal cancer ranks third in incidence, with more than 1.93 million new cases, and second in mortality, with 935,173 deaths in 2020 [2,3]. Approximately, 50% of patients exhibit a metastatic profile [4] and usually progress unfavorably due to their resistance to current therapies [5]. This worst prognosis is driven by several molecular mechanisms that play a critical role in the development, progression and chemoresistance of CRC [5,6].

Indeed, a malignant tumor is characterized by the acquisition of genetic and epigenetic alterations, which lead to uncontrolled pro-proliferative signaling, escape from tumor suppressors and immunosurveillance mechanisms, a resistance to cell death, deregulated energy metabolism, the induction of angiogenesis, invasion and metastatic dissemination. These properties contribute to the complexity of the neoplastic disease and are orchestrated by a network of cellular signaling pathways, such as Mitogen-activated protein kinase (MAPK), Phosphoinositide 3-kinase/Protein Kinase B (PI3K/AKT), Nuclear factor (NF)-κB, Janus kinase/Signal transducer and activator of transcription (JAK/STAT), Wingless-related integration site (Wnt)/β-catenin, Transforming growth factor-β (TGF)-β and Neurogenic locus notch homolog protein (Notch).

Furthermore, the development of solid tumors is not limited to these processes [7,8]. Indeed, numerous studies have highlighted the importance of interactions between precancerous and cancerous cells with their microenvironment during the different stages of their pathology [9]. The concept of the tumor microenvironment (TME) was first proposed by Virchow et al. [10], pointing out the relationship between inflammation and cancer [10]. Local inflammation at sites of solid malignancies results in the accumulation of a variety of cells that are closely associated with tumor growth promotion. Similar to most other solid tumors, colon cancer is infiltrated by various cells, such as CD4+ T cells, dendritic cells (DC), natural killer cells (NK), endothelial cells, endothelial progenitor cells (EPC), platelets, and mesenchymal stem cells (MSCs) [9]. This causes tumors have a complex multi-cellular ecosystem that facilitates the malignant potential of cancer development [11]. In addition, the metabolic changes that occur in TME can influence not only the biological activity of tumor cells, which become more aggressive and auto-sustained, but also the immune response against tumor cells, by either producing ineffective responses or polarizing the response toward pro-tumor activity [12]. Furthermore, immune cell composition within the TME can vary widely among patients with the same cancer type, suggesting that mapping the composition of immune infiltrates and their functional status within the TME is important for both diagnosis and therapeutic strategy development [13]. It has been reported that immune infiltrates cells can polarize into diverse cell types and exhibit a paradoxical role in the tumor tissue. Indeed, they can play both anti- and pro-tumor roles in TME orchestrating the regulation of cell polarity either by intrinsic and extrinsic factors [14].

Therefore, this review summarizes the cellular signaling pathways network linked to CRC, the paradoxical role of immune infiltrates cells within the TME and the cross talk between such different regulators.

## 2. The Role of the Signaling Pathways in the Carcinogenesis of CRC

A plethora of cellular signaling pathways that regulate multiple steps of tumor progression are involved in CRC development [15,16]. Among them, (Wnt)/β-catenin, MAPK, Notch, PI3K/AKT, NF-κB, TGF-β and JAK/STAT are key players of important roles in CRC malignancy [16,17].

### 2.1. The Wingless/Integrated (Wnt)/β-Catenin Signaling Pathway

Wnt/β-catenin signaling is involved in cell differentiation both during embryogenesis and during adult tissue homeostasis, in cell–cell communication system and in the regulation of cell proliferation [18,19]. The aberrant activation of this pathway induces the accumulation of β-catenin in the nucleus, leading to the upregulation of many cell cycle-related proteins, such as cellular myelocytomatosis (c-Myc) and Cyclin D-1, which promote carcinogenesis [20]. Mechanically, without active stimuli, β-catenin would not accumulate in the cytoplasm based on the fact that the adenomatous polyposis coli/Axin/Glycogen synthase kinase-3 (APC/Axin/GSK-3) complex proceeds to its degradation via the ubiquitin–proteasome pathway. Indeed, in the presence of a Wnt ligand, casein kinase 1 (CK1) phosphorylates β-catenin at Ser45. This event primes β-catenin for subsequent phosphorylation by GSK-3β at Ser33, Ser37, and Thr41 residues [21]. The phosphorylation and the inhibition of GSK3β ensure an increase in the concentration of cytosolic β-catenin [22] (Figure 1A). However, unphosphorylated β-catenin accumulates in the cytoplasm and then migrates to the nucleus where it binds to the lymphoid enhanced factor-1/T-cell factor 4 (LEF-1/TCF4) and other co-regulators in order to promote the transcription of target genes such as Jun, c-Myc, and Cyclin D-1 in a tissue-specific manner [19,23].

Wnt/β-catenin plays an important role in the pathogenesis of CRC [24]. A large sequencing project conducted on 1134 CRC samples identified diverse alterations in core WNT regulators; this mainly included the loss-of-function mutations in APC, which are very frequent among CRC patients. In addition, it has been demonstrated that more than 96% of CRC patients showed WNT activation [25]. Wnt/β-catenin is considered to be the most vital pathway accelerating the process of epithelial–mesenchymal transition (EMT) and enhancing the metastatic properties of colon cancer [26]. The increase in the nuclear β-catenin levels is thought to be a hallmark of aggressive CRC, leading to the activation of Wnt-related targets, including c-myc, cyclin D1, matrix metallopeptidase (MMP) 2, and MMP 9, thereby increasing the proliferation, invasion, and migration potential of cells [27,28,29]. In addition, the analysis of 155 colorectal cancer tissues highlighted a correlation between the high β-catenin expression and the reduction in CD8+ T-cell infiltration. Mechanistically, β-catenin can regulate the c-c chemokine ligand (CCL) 4 expression to recruit CD103+ dendritic cells to enable CD8+ T cell activation [30]. The stabilization of β-catenin results in the tumorigenic phenotypes often observed in CRC. For this reason, the stabilized β-catenin remains the most promising target for curing the disease [31] (Figure 1B).

In contrast, there are cases in which WNT signaling exerts anticancer effects (Figure 1C). Indeed, β-catenin signaling plays an important role in the normal differentiation of myeloid progenitors [32]. It has been reported that the down-regulation of β-catenin is critical for the accumulation of myeloid-derived suppressor cells (MDSC), which suppresses T-cell responses and promotes tumor proliferation [33]. The dysregulation of this signaling pathway in myeloid cells has been implicated in promoting polymorphonuclear (PMN)-MDSC expansion in cancer [34]. Similarly, it has been demonstrated that the phospholipase Cγ2 (PLCγ2)-β-catenin axis inhibits the accumulation and suppressive phenotype of PMN-MDSCs in mouse models of lung carcinoma and melanoma [35]. In addition, it has been reported that the antagonist of the Wnt-β-catenin pathway supports tumor progression by creating an immune suppressive environment in which tumor cells can grow unabated [33]. Additionally, β-catenin signaling is important in maintaining CD8+ T cell stemness and central memory responses, as well as enabling T cell tumor infiltration [36]. Consequently, β-catenin signaling in the immune system may play an important role in multiple ways; on one hand, as a negative regulator of PMN-MDSCs, and on the other hand, as a positive regulator of T cell function [37].

### 2.2. The Mitogen-Activated Protein Kinase (MAPK) Pathway

MAPK are ubiquitous signal transduction pathways that regulate all aspects of cell functions and are frequently altered in disease [38]. They are one of the most conserved signal transduction pathways, and have critical functions in cell proliferation, differentiation, death, and embryogenesis [39]. In the MAPK pathway, there are different phosphorylation cascades that modulate several series of vital processes. Each cascade is initiated by a specific extracellular stimulus and leads to the activation of particular MAPK kinase kinase (MAPKKK) and MAPK kinase (MAPKK). In a simplified model, the presence of mitogens and growth factors promotes the activation and dimerization of a canonical receptor tyrosine kinase subsequent to the activation of the small GTPases (Ras, Rac, RHO, or RAP). This induces the activation of MAPKKK (Raf or MEKs) and leads then to the stimulation of MAPKK (MEK1/2, MKK3/6, or MKK6/7). Finally, MAPKK activates the MAPK, including ERK1/2, p38, or JNK, by dual phosphorylation in order to interact and activate many transcription factors [39] (Figure 2A).

MAPK pathways play a critical role in different aspects of tumorigenesis, such as tumor growth, apoptosis, angiogenesis, invasion, metastasis, and drug resistance [39]. Dysregulated epidermal growth factor receptor (EGFR)/MAPK signaling pathway plays an oncogenic role in the initiation and development of CRC [40] (Figure 2B).

It has been reported that the long non-coding RNA H19 increases the migration and invasion of CRC cells by activating the RAS-MAPK signaling pathway, one of the most frequent carcinogenic events in human cancer [41]. Moreover, p38α MAPK signaling is a mediator of resistance in various agents in CRC patients, and it may also acquire an oncogenic role involving cancer related-processes, such as cell metabolism, invasion, inflammation, and angiogenesis [42]. In addition, the activation of the activator protein-1 (AP-1) by JNK and the nuclear factor-kappa B (NF-κB) by p38-MAPK promotes the invasion of human colon cancer cells [43]. For these reasons, MAPK pathways are mostly the target of cancer therapy.

The MAPK plays a crucial role in both cell proliferation and cell death [42]. The sustained activation of ERK1/2 promotes colon cancer cell death, and is induced by some anti-tumor compounds. In accordance, we previously reported that Lebein, a heterodimeric disintegrin isolated from *Macrovipera lebetina* snake venom, significantly inhibited the viability of LS174 colon cancer cells and induced their apoptosis by triggering the activation of the MAPK ERK1/2 pathway through the induction of reactive oxygen species (ROS) [44]. The halogenated monoterpene Mertensene from the red alga *Pterocladiella capillacea* induces G2/M cell cycle arrest and the caspase-dependent apoptosis of the human colon adenocarcinoma HT29 cell line through the modulation of intracellular ROS levels linked to the activation of the ERK1/2 in HT29 cells [45]. Fraxetin, a natural compound extracted from Fraxinus spp, induced apoptotic cell death in HT29 and HCT116 cells through mitochondria dysfunction associated with ROS induction, the modulation of ERK1/2, JNK, and P38 signaling pathways [46] (Figure 2C). JNKs have been also shown to play a role in apoptotic and non-apoptotic programmed cell death mechanisms. JNKs can either induce or inhibit cell death by stimulating the expression of specific genes and by modulating the activities of pro- and anti-apoptotic proteins through phosphorylation events [47]. It has been reported that treatment of HCT116 and HT29 colon cancer cells with a natural naphthoquinone 2-methoxy-6-acetyl-7-methyljuglone (MAM), isolated from *Polygonum cuspidatu*, validated the role of JNK in MAM-induced necroptosis, marked by mitochondrial depolarization, ATP depletion, and an increased production of mitochondrial ROS [48]. In addition, alantolactone, a plant-derived sesquiterpene lactone, showed anti-proliferative and pro-apoptotic effects in HCT116 colon cancer cells through activating the MAPK-JNK/c-Jun signaling pathway [49].

In addition, a synthetic alkaloid Lappaconitine hydrochloride exhibited antitumor activity in CRC HCT-116 cells by inducing their apoptosis through mitochondrial and MAPK signaling pathways [50].

### 2.3. The Neurogenic Locus Notch Homolog Signaling (Notch) Pathway

The Notch signaling pathway modulated a series of fundamental cellular functions, including cell fate decision, the maintenance of stemness, proliferation, and apoptosis [51]. There are four receptors: Notch-1, Notch-2, Notch-3, and Notch-4 of the notch pathway, which are normally activated by interacting with ligands such as Delta-like and Jagged [52]. In total, there are three delta-like (Dl) ligands (Dll1, Dll3, and Dll4) and two Jagged (Jag) ligands (Jag1 and Jag2) [52,53]. In the canonical Notch pathway, ligand–receptor interaction results in a cascade of proteolytic cleavages, first mediated by metalloproteases and second by γ-secretase activity. These cleavage steps result in the release of a constitutively active intracytoplasmic nick (ICN) fragment, which is then translocated to the nucleus where it associates with CSL and MAML as part of a larger transcriptional complex [52] (Figure 3A). The precise signaling differences between the Notch receptors and Notch ligand pairs are unknown. While they all seem to go through the same pathway, there is evidence that different receptor–ligand parings yield distinct biological outcomes [54]. In addition to their role in cellular functions, the Notch signaling plays a crucial role in many aspects of cancer biology, either oncogenic or tumor-suppressive [55]. This pathway is aberrantly activated in many cancers, including CRC. Recently, the analysis of the Notch signaling pathway of 1116 CRC patients in East China highlighted that one locus at MINAR1 out of 133 genes is significantly associated with overall survival. This study proves that the Notch pathway plays a crucial role in the progression of CRC, likely affecting patient survival [56]. Moreover, in CRC patients, it has been reported that mutations in the Notch pathway components can activate anti-tumor immune responses, which are characterized by the up-regulation of checkpoint molecules [57]. Previously, it has been shown that the dysregulation of the Notch pathway is linked to the pathogenesis of CRC and plays an oncogenic role in CRC development and progression [58,59]. Indeed, the upregulation of Jagged-1, mediated by β-catenin, increases Notch-1 expression [60], which has been correlated with progression, tumor grade, and metastasis in CRC. This could be related to the inhibition of apoptosis promoted by Notch-1 [61] and induced invasiveness through the activation of several pro-oncogenic factors, including CD44, Cyclin D1 (CCND1), and BCL2 Apoptosis Regulator [62]. Notch-1 and Notch-2 have been associated with opposite clinical outcomes in CRC patients. Indeed, the increased expression of Notch-1 predicted a poor overall survival, while, on the contrary, the reduced expression pattern of Notch-2 was linked to a worse one [63]. In CRC pathogenesis, Notch2 overexpression may activate the GATA3/IL-4 pathway, which subsequently promotes the polarization of the tumor-associated macrophages toward an M2 phenotype and then enhances EMT [64]. Notch2 has also been shown to mediate stemness promotion and chemoresistance in CRC cells [65]. Notch-3 is also upregulated in metastatic CRC and may regulate CRC-related tumorigenesis [66]. The dysregulation of Notch-3, as well as Jagged-1 and Dll-4, is associated with a more aggressive phenotype in xenografts of CRC cells in vivo [66]. In addition, Dll-1 can increase Wnt and TGF-β by binding to Smad2/3 and Tcf-4, and overexpressing the CTGF gene, which regulates colon carcinogenesis independently of Notch signaling [58,67]. In addition, the dysregulation of the Notch pathway in the tumor skews the local cytokine composition, shaping the immunological landscape and affecting tumor growth, progression, and metastasis [68] (Figure 3B).

The research around the role of Notch signaling in tumorigenesis has focused on its role as an oncogene. Interestingly, Notch activity has been associated with both oncogenic and tumor suppressor functions, which depended on the cellular context and the nature of the induced response [55,69,70]. Indeed, various studies in different cancers have revealed a clear link between the loss of Notch activity and carcinogenesis, suggesting that Notch plays an important tumor suppressor function in certain tissues [55]. The role of Notch signaling as a tumor suppressor has also been highlighted in breast, prostate, liver, lung and skin cancers [71]. These studies suggested that Notch plays a role in solid tumors through the defective activation of signaling pathways. Furthermore, the cellular outcome of this aberrant Notch activity is highly dependent on contextual cues, such as interactions with the tumor microenvironment and crosstalk with other signaling pathways [71]. The anti-tumor role of the Notch signaling pathway in CRC has not yet been clearly illustrated. Previously, it has been reported that in a total of 146 colorectal cancer samples, Notch2 and JAG1 expression levels were associated with patients survival [72] (Figure 3C). In addition, it has been demonstrated that the higher levels of infiltration of CD4+ T cells, macrophages, neutrophils, and dendritic cells were positively correlated to the expression of Notch receptors in patients with gastric cancer [73].

### 2.4. The Phosphoinositide 3-Kinase (PI3K)/Protein Kinase B (AKT) Signaling Pathway

PI3K/AKT signaling pathway plays a pivotal role in many biological and cellular processes, such as cell proliferation, growth, invasion, migration, and angiogenesis [74]. PI3K, a member of the lipid kinase family, is composed of a catalytic domain (p110) and a regulatory domain (p85). The activation of PI3K could catalyze the phosphorylation of phosphatidylinositol (PI) at the 3′-position of the inositol ring. The phosphorylated products have a critical influence on cellular functions, such as the enhancement of cell migration by PIP3, and the regulation of B cell activation and insulin sensitivity by the PI 3,4-bisphosphate. The PIP3 activates PDK1, which phosphorylates the serine/threonine kinase AKT at Thr308. AKT can also be phosphorylated and activated by PDK2 at Ser473. Activated AKT regulates cell proliferation, differentiation, migration, and apoptosis by activating or inhibiting downstream target proteins, such as Bad, Caspase 9, NF-κB, GSK-3, FOXO3, p21, p53, and FOXO1 (Figure 4). Additionally, AKT activates the mTOR pathway, thereby regulating cell growth through the modulation of the expression of cyclin D1 and p53. AKT boosts cell survival by inactivating the pro-apoptotic factor Bad and the transcription factor FKHR family [74]. The overexpression of PI3K/AKT/mTOR signaling, which is linked to the regulation of distinct oncogenic mechanisms, has been reported in various forms of cancers, especially in colorectal cancers (CRC). In addition, it plays a significant role in acquiring drug resistance, as well as in the metastatic initiation events of CRCs [75,76].

Due to the significant roles of this signaling pathway in the initiation and progression events of CRC, it is consistently recognized as a striking therapeutic target. PTEN, a tumor suppressor protein, possesses alkaline phosphatase and protein phosphatase activities, and can block PI3K/AKT signaling via the dephosphorylation of PIP3 to PIP2. In addition, Carboxyl-terminal modulator protein (CTMP) could block the activation of downstream signaling pathways by inhibiting AKT phosphorylation. The protein phosphatase 2A (PP2A) inhibits the activation of AKT through its dephosphorylation at Thr308 and Ser473 residues [74].

However, a study by Nogueira et al. [77], along with other research [78,79], reported upon a function of AKT and highlighted that it is not a single function kinase but, under certain conditions, can facilitate rather than inhibit cell death; this is via an increase in the reactive oxygen species and through suppressing antioxidant enzymes [77] (Figure 4). In accordance, our team previously reported that Mertensene, a halogenated monoterpene isolated from the red alga *Pterocladiella capillacea*, inhibited the viability of HT29 colon cancer cells and induced their apoptosis by triggering the activation of AKT [45]. The PI3K/AKT/mTOR pathway is also a critical regulator of cell autophagy. It has been reported that this process is involved in promoting cancer and participates in regulating the balance between the tumor and its microenvironment. Furthermore, the role of autophagy in cancer seems paradoxical due to its dual function as a survival or suppressor mechanism for tumor cells [80]. Indeed, it has been shown that targeting PI3K/AKT/mTOR-mediated autophagy is a double-edged sword in cancer. On the other hand, autophagy activation, by modulating the PI3K/AKT/mTOR pathway, increases the drug sensitivity of certain types of tumors and avoids drug resistance [81].

### 2.5. Nuclear Factor-kappaB (NF-κB) Signaling Pathway

NF-κB is one of the major signaling pathways involved in physiological and pathological conditions. It controls the expression of more than 400 genes, leading to their regulatory effects on different mechanisms, including immune response, inflammation, cell migration, apoptosis, and differentiation [82,83]. Indeed, extracellular stimuli, such as bacteria, viruses, cytokines, oncogenic molecules, chemo/radiotherapy, and cell surface receptors, including Toll-like receptors (TLRs), T/B cell receptors, and the interaction of tumor necrosis factor receptors (TNFR) with their specific ligands, cause the upregulation of the IκB kinase (IKK) complex [84]. The IKK complex phosphorylates p65/p50-bound IκB at serine residues -32 and -36. Phosphorylated IκB is degraded by the ubiquitin–proteasome pathway, thereby activating NFκB. The activated NF-κB is translocated to the nucleus, where it binds to the enhancer element of the immunoglobulin kappa light-chain of activated B cells (κB sites), thus triggering the expression of downstream genes that potentially leads to inflammation and the promotion of cancer development/progression [84,85,86,87] (Figure 5A). NF-κB exhibited an alternative pathway, initiated by ligands such as the cluster of differentiation (CD)-40, B-cell activating factor (BAFF), and lymphotoxin-β receptor (LTBR), and that includes RelB/p100 subunits of NF-κB, IKKα homo-dimer, and NF-κB-inducing kinase (NIK) [87].

The aberrant regulation of NF-κB is frequently reported in tumor cells and it is considered to be a poor prognostic marker in patients with CRC [88]. It has been reported that NF-κB suppression can induce apoptotic cell death in CRC cells [89]. In line with this, the activation of NF-κB signaling plays a significant role in the tumorigenesis process via the regulation of downstream NF-κB gene products linked to cell growth, inflammation, metastasis, angiogenesis and drug resistance in CRC cells [87] (Figure 5B). NF-κB inhibits apoptosis by up-regulating the expression of anti-apoptotic genes, including B-cell lymphoma-extra large (Bcl-xL), the Bcl-2-related gene (A1/BFL1), cellular inhibitors of apoptosis (cIAPs), and caspase-8/FAS-associated death domain-like IL-1beta-converting enzyme inhibitory protein (c-FLIP) [87,90]. It has been shown that NF-κB enhances the expression of various invasion-related genes, including MMPs, urokinase-type plasminogen activator (uPA), Vascular cell adhesion molecule 1 (VCAM-1), endothelial leukocyte adhesion molecule 1 (ELAM-1), Intercellular Adhesion Molecule 1 (ICAM-1), iNOS and COX2. In addition, it is involved in cancer-associated extracellular matrix (ECM) degradation [87,89]. Furthermore, the activation of NF-κB in response to chemotherapy reduces drug efficacy by decreasing the tumor cell chemosensitivity and cell death. Thus, the co-administration of chemo agents with NF-κB inhibitors can enhance chemosensitivity in CRC cells [91,92].

As described above, the NF-κB pathway plays a key role in supporting tumorigenesis, progression, and the chemoresistance of tumor cells. However, the role of the NF-κB signaling pathway in the induction of anti-tumor host immunity remains unclear. Recently, single-cell transcriptomics analyses have explored the molecular pathways that regulate the maturation of intratumoral conventional type 1 dendritic cells (cDC1s), which is critical for antitumor immunity, and have demonstrated the dynamic reprogramming of tumor-infiltrating cDC1s by NF-κB and IFN signaling pathways [93]. Moreover, the inactivation of NF-κB or IFN regulatory factor 1 (IRF1) in cDC1s, resulting in the impaired expression of IFN-γ-responsive genes, has been shown to lead to the ineffective recruitment and activation of antitumor CD8+ T cells [93] (Figure 5C). This study highlights the important role of NF-κB signaling in the polarization of conventional dendritic cells. Therefore, in cDC1, this pathway may represent an important focal point for the development of new diagnostic and therapeutic approaches in order to improve cancer immunotherapy.

Continuing in the discussion of the important role of NF-κB as an anti-tumor pathway, a recent study demonstrated that NF-κB-inducing kinase (NIK), which is known to be a mediator of noncanonical NF-κB activation, is a pivotal regulator of T cell metabolism and antitumor immunity [94]. Indeed, the transgenic expression of NIK, using an inducible system, strongly promoted the antitumor immunity associated with an increased glycolytic metabolism and the effector functions of CD8+ T cells (Figure 5C). This result was confirmed in a preclinical model of cancer therapy and proves that NIK expression rendered PD1+Tim3+ tumor-infiltrating CD8+ T cells competent for IFN-γ production [94]. In line with this, previously it has been reported that NIK plays a crucial role in mediating effector T cell function [95,96]. All these observations suggest that NIK regulates T cell function via both noncanonical NF-κB-dependent and independent mechanisms.

### 2.6. The Transforming Growth Factor-β (TGF-β) Signaling Pathway

The TGF-β signaling pathway is critical in many biological processes. It regulates cell growth, differentiation, apoptosis, cell motility, extracellular matrix production, epithelial–mesenchymal transition (EMT), angiogenesis, and cellular immunity [97,98,99]. There are three receptors in the TGF-β signaling pathway: Transforming growth factor-β receptor (TGFBR)-1, TGFBR2, and TGFBR3, and three ligand isoforms: TGF-β1, TGF-β2, and TGF-β3 [100,101]. The cascade initiates upon binding of the TGF-β ligand to the TGFBR2, inducing the formation of a heterotetrametric complex of TGFBR2 and TGFBR1 [100]. Subsequently, the TGFBR2 kinase domain activates TGFBR1 by phosphorylation, which, in turn, phosphorylates the suppressor of Mother Against Decapentaplegic (SMAD)2/3, which can be assembled into complexes with SMAD4 and then translocated to the nucleus where they can regulate the expression of target genes [102] (Figure 6A). The negative regulation of this pathway is assured by SMAD7, which can compete with SMAD2/3 for the catalytic site of phosphorylated or activated TGFBR1 and, thereby, inhibit the phosphorylation of SMAD2/3 [102]. The deregulation of this pathway is associated with many diseases, such as cancer [103]. Indeed, the aberrant activation of this signaling pathway promotes a variety of tumors, including hepatocellular carcinoma, pancreatic cancer, esophageal cancer, gastric cancer and CRC [99,104]. In tumor, the TGF-β signaling pathway exhibits a paradoxical role that is known as the “TGF-β paradox” [97] (Figure 6). On the one hand, TGF-β signaling could have a tumor suppressor function by inhibiting cell proliferation and stimulating cell differentiation in the early stages of cancer. On the other hand, in the late stages of cancer, it induces tumor progression and metastasis [97]. In CRC, its role also remains controversial. Indeed, it can promote or suppress the growth of colon cancer cells depending on the microenvironment [105]. Under normal conditions, TGF-β can induce the arrest of the cell cycle in G1 by increasing the expression of the cyclin-dependent kinase (CDK)4/6 inhibitor p15 [106] and by suppressing the multi-functional oncogene c-Myc [107]. In fact, during the early stages, TGF-β acts as a “tumor suppressor” and inhibits the proliferation of the tumor cells (Figure 6C). Previously, it has been reported that a bis-benzylisoquinoline alkaloid Tetrandrine inhibited the proliferation of colon cancer HCT116 cells and induced their apoptosis by increasing the mRNA and protein levels of TGF-β1. The upregulation of TGF-β1 decreases the phosphorylation of PTEN, thereby inactivating PI3K/Akt signaling [105]. In addition, it has been reported that TGF-β abrogated the epithelial–mesenchymal transition, and inhibited the invasion and migration of CRC cells, inducing the N-Myc downstream-regulated gene 2 (NDRG2) [108].

However, colon cancer cells can overcome the tumor-suppressing effects of the TGF-β pathway by inducing the loss of Smad proteins and deregulating the TGF-β type II receptor-mediated cell cycle [109]. Moreover, even if the TGF-β signaling contributes as a promoter of the tumor, its role in carcinogenesis is still complex. The ability of TGF-β to act either as a tumor suppressor or as an oncogenic agent is determined by the cell-to-cell communications and by the tumor stage [110]. Indeed, during late stages, TGF-β supports tumor cell proliferation, invasion, and metastasis (Figure 6B). The change in TGF-β expression and in the cellular responses tips the balance in favor of oncogenic activities by inducing the EMT, which is mediated by Fibronectin, Twist, and Snail, and accelerates tumor invasion and metastasis [108,111,112]. Previously, it has been reported that TGF-β1 increased the progression of colon cancer by upregulating the expression of Human Cripto-1 (CR-1) linked to tumorigenesis [113,114]. The overexpression of TGF-β has been highlighted in different tumors [97,115]. Moreover, this signaling pathway enhances angiogenesis via the up-regulation of the Vascular Endothelial Growth Factor (VEGF) [116] through Smad-dependent pathways [97,117,118]. Under normal conditions, TGF-β plays a key role in controlling immune responses. It promotes the differentiation of regulatory T cells and Th17 cells [119,120,121]. TGF-β suppresses the immune system by inhibiting NK-cell activity [122], decreasing cytokine production, inhibiting dendritic cell maturation [123], and altering T-cell cytotoxic properties [124]. All these properties suggest that TGF-β exhibited a facilitative and a direct role in tumor progression as it directly suppresses the immune system and allows tumor cells to acquire properties that help to evade the immune system [97].

### 2.7. The Janus Kinase/Signal Transducer and Activator of Transcription (JAK/STAT) Signaling Pathway

The JAK/STAT signal transduction pathway is the common signaling pathway in which many growth factors and cytokines can transmit signals in cells [125,126]. The JAK family comprises four members: JAK1, JAK2, JAK3, and TYK2 [127], which will act through seven STAT family members, STAT1, STAT2, STAT3, STAT4, STAT5a, STAT5b, and STAT6 [126]. The cascade begins upon the binding of cytokines, inducing the receptor oligomerization and then leading to the recruitment of related JAKs. JAK activation induces the phosphorylation of tyrosine in the intracellular domain of the receptors. Once activated, JAKs serve as docking sites for STAT. Then, STATs dissociate from the receptor to form homodimers or heterodimers, and translocate to the nucleus to initiate the transcription of a repertoire of target genes [128] (Figure 7A). The JAK/STAT signal transduction pathway can promote the expression of several downstream genes that contribute to the many biological processes that are involved in immune function and cell growth [129]. In normal conditions, it plays an important role in the growth and development of the body. Under pathological conditions, the activation of the JAK/STAT pathway mediates the proliferation, differentiation, and migration of malignant tumor cells [130] (Figure 7B). During the early and late stages of CRC, JAK/STAT signaling is considered to be a clinical predictor and a prognosis marker for diagnosis, making this pathway as a target for therapeutic intervention [130,131,132]. Previously, it has been reported that the JAK/STAT pathway is activated in colon cells [133]. Case-control studies, using more than 1550 patients with colon cancer and 750 cases of rectal cancer, demonstrated that JAK2, SOCS2, STAT1, STAT3, STAT5A, STAT5B, and STAT6 were associated with colon cancer, and that STAT3, STAT4, STAT6, and TYK2 were linked to rectal cancer [129]. Thus, the rapid transduction of extracellular signals by the JAK/STAT pathway to the nucleus plays a pivotal role in the activation of oncogenes and the negative modulation of tumor suppressor genes in colon cancer [130]. Moreover, JAK/STAT can activate angiogenesis through the overexpression of many downstream growth factors, such as VEGF, insulin-like growth factor-1 (IGF-1), and MMP [134] (Figure 7B). JAK/STAT also plays regulatory roles in the inflammatory response, glycolysis, and epithelial–mesenchymal transition [130,135,136]. It has been reported that, depending on the cellular context, IFN/STAT signaling could mediate tumor cell growth, metastasis, and chemo- and radio-resistance to therapies [137,138]. Various studies have shown that the hyperactivation of STAT3 enhances the expression of its target genes, leading to an increase in tumor cell migration and proliferation, and thus contributing to colorectal carcinogenesis [139,140]. For this reason, the deletion of the hotspot mutation region in the DNA-binding domain of STAT3induced colon cancer cell growth and progression due to genome-wide changes in the transcription of STAT3-target genes [140]. In line with this, the analysis of the association of JAK-1 and STAT-3 protein levels with the clinicopathological parameters of patients with colon cancer has demonstrated that the expression of these two proteins is associated with the clinical stage of the pathology [130]. Indeed, STAT3 was shown to also inhibit cell apoptosis through down-regulating the apoptotic protein Bcl-xl [141]. In addition, based on its role in the regulation of the immune response, STAT3 has been reported to regulate the differentiation of Th17 cells [142]. In addition, it promoted the immunosuppression of tumor-associated macrophages and myeloid-derived suppressor cells [143,144] (Figure 7B).

Like several other signaling pathways, the JAK/STAT pathway could exhibit both a pro- or anti-tumor role. The complexity of these effects depends on several factors, such as the stage of the pathology and/or the cellular context. For example, IFN/STAT signaling has been known as an anti-tumorigenic pathway that regulates many effectors, such as caspases [145,146,147], cyclin-dependent kinase inhibitors [148], the anti-apoptotic protein BCL2 [149] and the IRF1/p53 pathway [150]. However, there is increasing evidence that IFN/STAT singling also contributes to tumor initiation and dissemination [137]. Furthermore, the role of STAT1 and STAT3 in CRC development and progression is controversial. It has been demonstrated in vivo that there are two opposite cell growth behaviors based on the STAT1/3 expression patterns (Figure 7). Indeed, a low STAT1/high STAT3 ratio highlighted a faster tumor growth in xenografts compared to high STAT1 and low STAT3 expressions, which were characterized in contrast slower ones. Interestingly, the simultaneous absence of nuclear STAT1 and STAT3 expression was associated with a reduction in the median survival by ≥33 months [132]. In CRC, the role of STAT3 is paradoxical. Indeed, as explained above, STAT3 activation can upregulate the expression of MMP, which, in turn, could favor cancer cell invasion and metastasis [151,152]. However, it has been shown that STAT3 activity may also exert limiting effects on colonic carcinoma development in murine models, as well as suppress tumor cell invasiveness [153,154]. In line with this, a study on CRC tissue microarrays showed that the expression and/or activation of STAT3 indicated a favorable clinical prognosis outcome [154] (Figure 7C). This double role may be explained by the presence of two splicing isoforms of STAT3, STAT3α and STAT3β, which have different functions in the regulation of tumors. STAT3α activation is believed to promote tumor initiation, while STAT3β is believed to inhibit the occurrence of cancer and is considered to be an effective tumor suppressor [128,155,156].

Another component of the JAK/STAT pathway that has a contradictory role is STAT1. On the one hand, the aberrant expression of STAT1 is found in tumor cells [157]. Moreover, STAT1 has a strong association with indoleamine-2,3-dioxygenase-1 expression in Paneth cells in the stem cells of intestinal crypts and tumors, and with subsequent immune escape in CRC [158] (Figure 7B). On the other hand, based on its anti-proliferative and pro-apoptotic effects, STAT1 is generally regarded to have tumor-suppressive functions and in CRC, its activity was shown to be associated with a favorable prognosis [154,159] (Figure 7C). In animal models, the loss of STAT1 expression promotes the development of CRC [131]. It has been demonstrated that STAT1 deficiency promotes rapid and extensive intestinal damage, leading to increased proliferation in the early stages of induced tumor formation and reduced apoptosis in advanced tumors [131]. This study supported the important role of STAT1 signaling during the development of cancer cells in vivo. Thus, STAT1 promoted cell apoptosis by activating the apoptotic caspases 1, 3, and 11 precursors, and by interacting with the p53 protein. Furthermore, STAT1 can also induce Fas, Bcl-2, and Bcl-X gene expression [128]. In addition, a recent study highlighted that increased STAT1 expression in tumor cells was strongly indicative of an immunogenic microenvironment, characterized by significantly high expression levels of MHC class I and PD-L1, both on tumor and non-tumor cells. Furthermore, tumor-infiltrating lymphocytes (TILs) were also increased in the positive-STAT1 group [160].

Thus, even if the activation of these signaling pathways and their link to CRC is well established, the interconnection between their different components, their dual functions and also the intercellular communications between tumor cells and the TME are still not fully understood; this contributes to the difficulty of achieving a successful therapy for CRC.

In accordance, in the next section, we will discuss how such crosstalk contributes to colorectal cancer development and progression, focusing on the double-edged role of some of the effectors regulating these processes. The comprehension of such mechanisms provides new opportunities for the development of efficient therapies for CRC.

## 3. The Role of Tumor Microenvironment in the Carcinogenesis of CRC

The communication between tumor cells and the tumor microenvironment (TME) is established through paracrine factors that trigger the activation of numerous signaling pathways [161]. Indeed, carcinogenesis in colorectal cancer (CRC) is critically influenced by the TME, and guided by a plethora of different cells and effectors.

### 3.1. Tumor-Associated Macrophages (TAM)

TAMs are one of the most abundant cell types that play a pivotal role in the pathogenesis of CRC [162]. However, the cross talk between cancer cells and macrophages in TME is complicated and the underlying mechanisms are still poorly elucidated.

TAMs are divided into the anti-tumorigenic “M1” phenotype and the pro-tumorigenic “M2” phenotype (Figure 8). Typically, in response to inflammatory stimuli, monocytes differentiate into activated pro-inflammatory M1 cells [163]. Mechanistic studies showed that the expression of phospholipase D4 (PLD4) in TAMs promotes the activation of M1 macrophages, resulting in an antitumor effect on colon cancer cells [164]. It has been confirmed that PKCα acts as a tumor suppressor through the MKK3/6-p38 MAPK signaling pathway to promote IL12/GM-CSF-mediated M1 polarization, and inhibited the growth of mouse colon cancer [165]. M1 macrophages exert their pro-inflammatory effect by the secretion of different cytokines, such as TNFα, IL-1β, and IL-6. They also secreted factors such as TNFα and ROS to both exert antitumor effects and stimulate cytotoxic T cell recruitment into tumors [166,167]. Moreover, M1 macrophages can exhibit an anti-tumor effect by phagocytosing tumor cells [168] (Figure 8A).

In contrast, the promotion of the M2 polarization of TAM is due to the secretion of EGF by the colon cancer cells through the EGFR/PI3K/AKT/mTOR pathway [169]. Wnt5a could also induce the M2 polarization of TAMs by regulating CaKMII-ERK1/2-STAT3 pathway-mediated IL-10 secretion, ultimately promoting the tumor growth and metastasis of CRC [170]. Recently, it has been reported that exosomal miR-106b induces M2 macrophage polarization by directly suppressing programmed cell death 4 (PDCD4) to activate the PI3Kγ/AKT/mTOR signaling pathway. The activated M2 macrophages enhance the ability of CRC cells to migrate, invade, and induce metastasis in vitro and in vivo [171] (Figure 8B). They are also linked to chemoresistance and drug-induced apoptosis inhibition via the secretion of IL6, which regulates the STAT3-miR-204 axis in CRC cells [172]. High levels of growth differentiation factor 15 (GDF15), which is produced by TAMs, also impair the chemosensitivity of tumor cells via enhancing fatty acids β-oxidation [173]. Additionally, TAMs express the programmed cell death protein 1 (PD-1), which inhibits phagocytosis and antitumor immunity [174]. In addition, it has been reported that TAM-derived CCL5 facilitates the immune escape of CRC cells via the p65/STAT3-CSN5-PD-L1 pathway [175]. Similarly, a recent study characterized a new macrophage subpopulation with a high level of PD-L1. This population was induced by tumor cells during macrophage infiltration and was associated with a poor prognosis [162,176]. Mechanistically, CRC-derived multiple sEV-miRNAs synergistically induced TAM M2-like polarization to promote the inhibition of CD8+ T lymphocytes by expressing PD-L1 through the PTEN/AKT and SOCS1/STAT1 signaling pathways; they thus contribute to immune escape and CRC progression [162]. This finding proves that inhibiting the secretion of specific miRNAs from CRC and targeting PD-L1 in TAMs may serve as novel methods for CRC treatment. Moreover, to suppress the function of CD8+T cells and DCs, and to stimulate the amplification of Treg cells, TAMs secrete IL-10 and TGF-β [177]. The production of TGF-β1 by M2 macrophages through the VEGF/VEGFR2 signaling pathway highlights the fact that M2-TAMs suppressed the anti-tumor immune response through paracrine and autocrine VEGF signaling via VEGFR2 in the tumor microenvironment of CRC [178].

### 3.2. Tumor-Associated Neutrophils (TANs)

TANs constitute a significant part of the tumor microenvironment as they play a substantial role in linking inflammation and cancer, and are also involved in tumor progression and metastasis [179]. For this reason, neutrophils could be considered to be one of the emerging targets in multiple cancers [180]. TANs are divided into two populations: the anti-tumorigenic “N1” phenotype and the pro-tumorigenic “N2” phenotype, which means also that TANs exhibit considerable plasticity and are capable of polarization into either tumor-suppressive or -supportive cells [163] (Figure 9). Previously, it has been reported that N1 neutrophils increase cytotoxicity and reduce the immunosuppressive activity of immune cells by producing TNFα, the intercellular adhesion molecule (ICAM)-1, ROS, and Fas, and by decreasing arginase expression (Figure 9A). In contrast, N2 neutrophils support tumor expansion by expressing arginase, MMP-9, and VEGF, and by reducing ROS production and the intra-tumoral recruitment of cytotoxic T lymphocytes [181] (Figure 9B). The plasticity of TANs is regulated by TGF-β and IFN-β signaling [182]. Indeed, TGFβ signaling functions as a regulator between the N1 and N2 phenotypes, and its inhibition induces an anti-tumoral N1 phenotype. Hence, the anti-TGF-β treatment increases the cytotoxicity of TANs, decreases the metastasis, and significantly increases the apoptosis of CRC cells by suppressing the activation of the PI3K/AKT pathway in TANs and TGF-β/Smad signaling in tumor cells [183]. Furthermore, TANs contribute to tumor invasion and angiogenesis, and promote tumor cell dissemination by capturing circulating tumor cells using neutrophil extracellular traps (NET) and promoting their migration to distant sites. NETs can activate toll-like receptor 9 on CRC cells, resulting in cellular growth, migration, and invasion via the activation of MAPK signaling [182]. Recently, it has been reported that IL-8 upregulation induces neutrophil enrichment and NET formation in KRAS-mutant tissues, which promote the growth of CRC cells [184]. Moreover, the neutrophil-to-lymphocyte ratio is a well-defined predictive marker for CRC patients, and a high ratio is considered to be a poor prognostic factor for this cancer [182]. Similarly, a retrospective study using a cohort of 354 CRC patients with stage I–III cancer revealed a strong relationship between dynamic changes in the neutrophil-to-lymphocyte ratio and overall survival [185].

All these data support that TANs exhibit contradictory roles in CRC with both pro-tumoral and anti-tumoral properties depending on the immunological context.

### 3.3. Cancer-Associated Fibroblasts (CAF)

Fibroblasts, commonly known as cancer-associated fibroblasts (CAFs), are the most abundant cells in the TME that influence tumor growth [186]. Nevertheless, the role of CAFs in colorectal cancer (CRC) development is not fully understood.

CAFs are a heterogeneous and plastic population. Different elements contribute to this heterogeneity, including the tissue type in which the tumor grows, the local paracrine environment, and the cell type of origin [187]. CAFs originate in their majority from the “activation” of local tissue fibroblasts via the action of tumor cell-secreted factors, such as TGFβ or platelet-derived growth factor (PDGF) [188]. Like other cells of the TME that display an enormous grade of plasticity, CAFs also have considerable plasticity and are capable of polarization [187]. The concept of “CAF polarization” highlights the functional heterogeneity of CAFs. They are divided into two functionally distinct subtypes: F1 and F2 polarized fibroblasts [187]. F1 represents CAF with antitumor effects, while F2 is CAF with tumor-promoting properties (Figure 10). However, the potential antitumor effects of CAFs are by far less studied compared to their tumor-promoting activity. Previously, it has been reported that primary fibroblasts, established from both normal and cancer tissues, can inhibit the proliferation of a panel of co-cultured cancer cells in vitro [189]. In breast cancer, the activation of Robo1 signaling by Slit2 from a stromal fibroblast prevents tumorigenesis, via the blocking of the PI3K/AKT/β-catenin pathway [190]. Similarly, it has been observed that fibroblast-derived Wnt3a could inhibit the growth of different patient-derived breast xenograft tumors [191]. A recent study demonstrated that Wnt induced a phenotypic switch in CAFs into non-aggressive cells and inhibited EMT in CRC [192]. Furthermore, the effects of CAFs on tumorigenesis appear to be less dependent on the instructive role of CAFs, but rather on the interacting compartments. For example, as described previously, Wnt/β-catenin signaling can be linked to the antitumoral effect of CAFs both in CRC and breast cancer [191,192]. In contrast, in CRC, it has been reported that WNT2 is selectively elevated in CAFs, leading to increased invasion and metastasis [193,194]. Similarly, as discussed above, one mechanism by which CAFs exert a tumor-suppressive effect is Slit2-induced Robo1 signaling, which is correlated positively with TGF-β [190]. However, TGF-β, for which CAFs are an important source, promotes tumor progression and metastasis [195] (Figure 10A).

It seems that CAFs can promote and suppress tumor formation, depending on their polarization profile in the local TME. The tumor-promoting activity of CAFs includes potent paracrine effects, which regulate the various cell types present in tumors [186,196] (Figure 10B). CAFs cooperate with tumor cells to promote the formation and maintenance of the tumor microenvironment by activating multiple signaling cascades, including the EGFR, JAK/STAT, TGF-β, and Wnt signaling pathways [197]. They promote an immunosuppressive microenvironment through the induction and accumulation of pro-tumoral macrophages [198]. However, the impact of the molecular crosstalk of tumor cells with CAFs and macrophages on monocyte recruitment and their phenotypic conversion is not fully depicted. It has been reported that colon fibroblasts and non-tumor cells recruit and dictate the fate of infiltrated monocytes towards a specific macrophage population, characterized by high CD163 expression and CCL2 production. Cytokine profiling revealed that CAFs produce M-CSF, IL6, IL8, and HGF. Moreover, macrophage/CAF/tumor cell co-cultures lead to the increased invasion of cancer cells [199]. CAFs can promote the polarization of macrophages toward the M2 phenotype, which contributes to the suppression of the functioning of natural killer cells through a synergistic mechanism. In addition, CAFs can also promote the adhesion of monocytes by up-regulating VCAM-1 expression in colorectal cancer cells. Thus, after VCAM-1 knocking-down in tumor cells or the depletion of macrophages, the pro-tumor effect of CAFs is partly abolished [200]. Recently, it has been demonstrated that IL1β, secreted by CRC cells, modifies surrounding normal fibroblasts in order to acquire protumorigenic characteristics and generate particularly chemoresistant cells [201]. Mechanistically, FGF-1/-3/FGFR4 signaling in cancer-associated fibroblasts promotes tumor progression in colon cancer through ERK and MMP-7 [202]. In addition, it has been reported that the constitutive activation of STAT3 in the CAFs of CRC promotes tumorigenesis [203]. Additionally, the concomitant activation of Wnt signaling and YAP/TAZ signaling coordinate to generate CAFs in CRC [204].

### 3.4. Tumor-Infiltrating Lymphocytes (TIL)

Immune responses play important and complex roles in immune surveillance and antitumor immunity during CRC progression [205]. TILs are located both within the tumor and in the peritumoral stroma. They are highly heterogeneous based on their cell-type compositions, gene expression profiles, and functional properties, which might contribute to the number of diverse responses to cancer immunotherapies [205]. TILs composed of NK cells, CD8+ cytotoxic T cells, and CD4+ helper T cells, are essential players in the defense against tumor cells [206]. Both CD8+ T cell and CD57+ NK cell infiltration, and Th1-type responses, are associated with a good prognosis in patients with colorectal cancer. However, the function of TILs can be controlled and inhibited by Treg cells [207]. Using the clustering analysis, Yang et al. identified eight T cell types from tumor tissues, including tumor Tregs, CD4+/CD8+ TRM T cells, CD4+/CD8+ effector memory T cells, Th17 cells, depleted CD8+ T cells, and CD8+ intraepithelial lymphocytes [208]. These T cell types likely represent the predominant tumor-infiltrating T cell subset in moderately differentiated CRC [208]. Treg cells in human colon tumor tissues express immunosuppressive molecules such as PD-1, cytotoxic T-lymphocyte-associated protein-4 (CTLA-4), T cell immunoglobulin and mucin domain-3 (TIM-3) and lymphocyte-activation gene 3 (LAG-3) [209]. In colon tumors, Treg cell depletion results in an increased accumulation of conventional T cells, including Th1-type T cells and IL-17A-producing T cells, suggesting that Treg cells not only regulate lymphocyte effector functions but also their recruitment to effector sites. Therefore, targeting Treg cells in antitumor immunotherapy can not only enhance the effector function of activated T cells, but also increase their numbers in tumors [210]. Of the TILs, helper T lymphocytes are considered to be the main players in tumor immunity that influence tumor progression. T helper type 1 (Th1), Th2, Th17, and regulatory T (Treg) cells have been identified as subtypes of helper T cells [210]. In TME, lymphoid cells exhibit immune-suppressive or stimulatory capacities. In the Th1/Th2 paradigm, cytotoxic T cell activity is supported by the Th1 lineage and M1 macrophages. In contrast, regulatory T lymphocytes, B-lymphocytes, and M2 macrophages are more closely related to tumor-promoting Th2 responses [211] (Figure 11). In healthy individuals, the Th1 and Th2 populations regulate each other. In patients with cancer, this balance shifts in favor of the Th2 cells. In CRC, it was reported that Th1 cell predominance is correlated with a good prognosis, whereas a high proportion of Th2 cells is associated with a worse one [207]. Indeed, M1 macrophages secrete cytokines such as IL-12, and can aid in the generation of Th1 adaptive immunity and impart a direct cytotoxic effect on tumor cells [212]. Recently, it has been reported that increased TGFβ in the tumor microenvironment represents a major mechanism for immune evasion, promoting T cell rejection and blocking the acquisition of a Th1 effector phenotype [213]. Interestingly, it has been demonstrated that exosomes derived from heat-stressed tumor cells possess a powerful capacity to convert immunosuppressive Tregs into Th17 cells via IL-6, which contributes to their potent antitumor effect [214]. In colon cancer, the immunoscore has extensively validated the prognostic significance of TILs. Based on this scoring system, the concept of “hot” (T-cell inflamed) and “cold” (non-T-cells inflamed) has emerged. Hot tumors are characterized by T cell infiltration and the molecular signatures of immune activation, whereas cold tumors show the striking features of T cell absence or exclusion [13]. Tumor intrinsic signaling pathways regulate T cell exclusion or their infiltration into tumors, which is crucial for developing novel therapeutic strategies against cancer. Local adaptive immunosuppression may be induced by involving the activation of various oncogenic pathways [215]. Recently, it has been reported that the downregulation of STAT1 induces immune escape in CRC, and that its upregulation implies a “hot” immunogenic microenvironment [160]. Furthermore, IFN-γ/JAK/STAT1 signaling stimulates PD-L1 expression in CRC cells [216]. The co-expression of the phosphorylated form of STAT1 and PD-L1 in CRC cells is strongly correlated with CD4- and CD8-positive TILs [217]. In addition, the intrinsic IFN type I (IFNα and IFNγ) signaling pathway is essential for the cytotoxic T lymphocyte (CTL) effector function in tumor suppression. Indeed, to evade host cancer immunosurveillance, human CRC may down-regulate the IFNAR1 on CTLs to impair the CTL effector function [218]. Another recent study has shown that TGF-β and PI3K signaling in CD4+ T cells (specifically Th17 cells) promotes the emergence of IL-22-producing Th17 cells and thereby tumorigenesis in mice [121]. An analysis of 155 colorectal cancer tissues highlighted that tumors with high β-catenin expression are characterized by a significant reduction in CD8+ T-cell infiltration. Mechanistically, β-catenin can regulate CCL4 expression to recruit CD103+ dendritic cells to enable CD8+ T cell activation [30]. Similarly, the immune escape in cancer is associated with the activation of various pathways, such as WNT–β-catenin, MAPK, JAK/STAT3, and NF- κB signaling [205]. The engagement of these pathways results in the production of cytokines and chemokines that ultimately mediate the exclusion of T cells from the TME, or the repression of factors that facilitate T cell recruitment.

### 3.5. Tumor-Associated Dendritic Cells (TADC)

TADCs play a key role in the orchestration of innate and adaptive antitumor immunity [219]. Therefore, dendritic cells (DCs) can strongly influence tumor progression and have a major impact on clinical outcomes in cancer patients [220]. They act as sentinels, detect tumor antigens, present them to CD8+ T-cells, and supply necessary signals for both the activation and suppression of CD8+ T-cells [221]. DCs play relevant roles in tumors by exerting both pro- and anti-tumorigenic functions depending on the local environment. The quantitative and functional impairment of DCs is widely observed in several cancer types, including CRC, and represents a tumor mechanism that is used by cancer cells to escape host immune surveillance [219]. Conventional DCs (cDCs) can be broadly divided into cDC1 and cDC2 populations (Figure 11), which arise through distinct pre-DC lineages [222]. Single-cell RNA sequencing, using CRC tissues, identifies three DC subsets: plasmacytoid DC (pDC), cDC2, and cDC1 cells, which are characterized by the high expression of HLA-DRs and the low expression of CD14, expressing specifically LILRA4/LILRB4, CD1C/FCER1A, and XCR1/ BATF3, respectively [223]. cDC1s are critical for generating anti-tumor T cell responses because cDC1s are capable of presenting tumor-associated antigens to CD8+ T cells and Th1 cells [221,224]. They produce type I and type III IFN and IL-12 at lower levels, and correlate with a favorable prognosis in cancer patients [224]. They are considered central to antitumor immunity and their presence in the tumor microenvironment is associated with improved outcomes in patients with cancer. Recently, it has been reported that the accumulation of CD8+ T cells indeed relies on BATF3-dependent cDC1s [225]. A direct correlation between Notch2 signaling, the infiltration of cDC1, and the association of the suppressed cDC1 signature with a poor prognosis in human CRC has also been shown. These findings reveal a critical role for Notch2-dependent cDC1s in preventing inflammation-associated transformation in the mouse model and tumor progression in human CRCs [226]. However, the role of cDC2s in tumors remains unclear. They exhibit substantial heterogeneity and they preferentially initiate CD4+ Tconv responses in a variety of immunological models [227]. Similarly, scRNA-seq and functional data in the TME demonstrate that two distinct populations of IRF4-dependent CD11b+ cDC2 are required for initiating the activation of antitumor CD4+ Tconv in vivo. cDC2 presented tumor-derived antigens to CD4+ Tconv, but failed to support antitumor CD4+ Tconv differentiation. However, Treg cell depletion enhanced their capacity to elicit strong CD4+ Tconv responses and ensure antitumor protection [228]. As reviewed, cDC2 expresses a large repertoire of pattern recognition receptors (PRRs) and pro- and anti-inflammatory cytokines, including IL-12. Mainly, they induce Th17 cell activation, but also Th1 cell, Th2 cell, Treg cell, and CD8+ T cell (cross-presentation) activation, which depends on the context and precise cDC2 subpopulation [224]. It has been reported that both cDC1 and cDC2 cells are programmed to differentiate into regulatory subsets upon their uptake of tumor antigens [229]. Thus, cDCs play an immunoregulatory role in the tumor microenvironment. The plasmacytoid dendritic cells (pDCs) can be stimulated to activate CD8+ T cells to generate powerful tumor antigen-specific CD8+ T cell responses [224,230]. The role that this DC subset plays in tolerogenic settings is poorly described, but correlates with a poor prognosis in cancer [224]. This is mainly due to the association that is made between the accumulation of pDC and the increase in Tregs cells, with the presence of TGF-β and/or IL10; these have an immunosuppressive phenotype, which results in a decreased overall survival rate in the patients [231]. Similarly, the analysis of a cohort of 64 gastric cancer patients without preoperative chemotherapy demonstrated an increased infiltration of BDCA2+ pDCs and Foxp3+ Tregs in tumors compared to normal ones. Indeed, BDCA2+ pDCs were positively associated with Foxp3+ Tregs [232], which were both considered to be immunosuppressive cells in the TME. Previously, it has been reported that pDCs can generate Tregs from naive CD4+ T cells, thereby contributing to tumor immune escape [233,234]. In addition, in ovarian cancers, they are essential for immunosuppression due to their expression of indoleamine 2,3-dioxygenase 1 (IDO1) and the inducible T cell costimulatory ligand (ICOSL) [235]. In contrast, it has been reported that higher densities of tumor-infiltrating pDCs are associated with better prognosis and the prolonged survival of patients with colon cancer. This effect is due to the co-localization of pDC and CD8+ T cells in tumor stroma, where activated pDCs may contribute to the stimulation of tumor-reactive CD8+ cytotoxic T cells [220]. These findings suggest that the TME plays a critical role in shaping the functional properties of infiltrating pDCs (tolerogenic vs activated). It has been shown that immunosuppressive DCs can be generated depending on the TME components, such as β-catenin, STAT3, TGF-β, IDO, endoplasmic reticulum (ER) stress, increased levels of lactate, VEGF, IL-10, TGF-β, prostaglandins, the accumulation of adenosine, and hypoxia [236,237].

## 4. Conclusions

Despite the progress made in the management of CRC, there are still many challenges to overcome in the development of antineoplastic treatments.

Through this review, we tried to actualize and compile what is known about the pivotal role of some signaling pathways in CRC development and progression, focusing on the paradoxical role of immune cells as beneficial or detrimental markers, and the crosstalk between tumor cells and the TME.

Hence, the double-edged sword function of the regulators that orchestrate tumor dissemination and the complex interactions that take place in the TME could explain the limited efficacy of therapeutic drugs and suggest that caution is applied before their administration to patients. Accordingly, an accurate staging of patients based on specific molecular groups, target identification and TME interactions could be a valuable tool to guide the optimal treatment.

Thus, modeling these complexities at the level of the individual patient and developing personalized anti-tumor approaches that block this crosstalk could rationalize the immunotherapy use and contribute not only to refining and to adapting treatment, but in improving the efficacy of therapies and bypassing anticancer treatment failure.

## Figures and Tables

**Figure 1 ijms-24-05600-f001:**
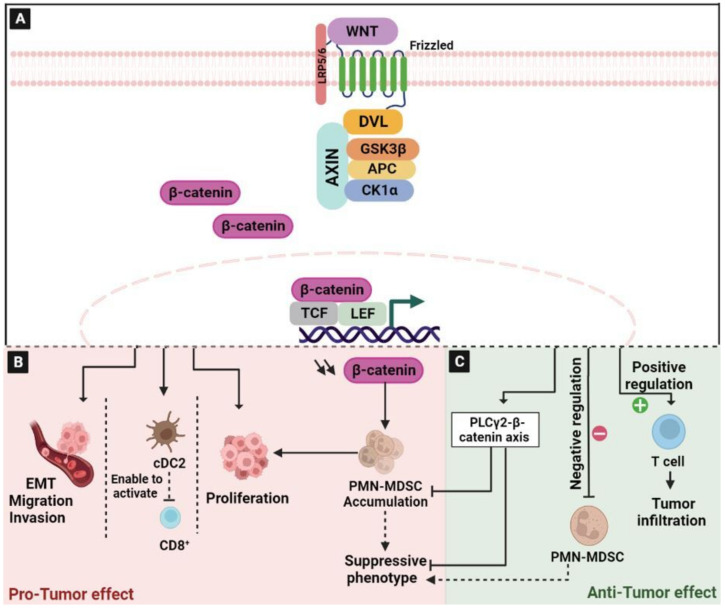
The dual function of Wnt/β-catenin signaling pathway in colorectal cancer. (**A**) Schematic representation of activated Wnt/β-catenin pathway: The ligation of WNT to their receptor Frizzled/LRP5/6-activated DVL and acted as a suppressor of GSK3β. This inhibition induces the accumulation of β-catenin in the cytoplasm. Subsequently, β-catenin is translocated into the nucleus and binds to TCF/LEF transcription factors, leading to the target genes’ transcription. (**B**) The wnt/β-catenin pathway increases the proliferative rate of tumor cells, accelerates the process of the EMT, and enhances the metastatic properties of cancer cells. In addition, β-catenin can regulate the CCL 4 expression to recruit cDC2 to enable CD8+ T cell activation. The down-regulation of β-catenin is critical for MDSC accumulation, which leads to the suppression of T-cell responses and promotes tumor proliferation. (**C**) β-catenin is a negative regulator of PMN-MDSCs and a positive regulator of T cell function. In addition, PLCγ2-β-catenin axis inhibits the accumulation and suppressive phenotype of PMN-MDSCs. Abbreviations: Wnt: Wingless/integrated, LRP5/6: Lipoprotein Receptor-related Protein 5/6, DVL: Dishevelled, GSK3β: Glycogen synthase kinase-3 beta, APC: Adenomatous Polyposis Coli, CK1α: Casein kinase-1 alpha, TCF: T-Cell Factor, LEF: Lymphoid Enhanced Factor, PMN-MDSC: Polymorphonuclear Myeloid-derived suppressor cells, EMT: Epithelial–Mesenchymal Transition, cDC2: Conventional Dendritic Cell 2, PLCγ2: Phospholipase Cγ2.

**Figure 2 ijms-24-05600-f002:**
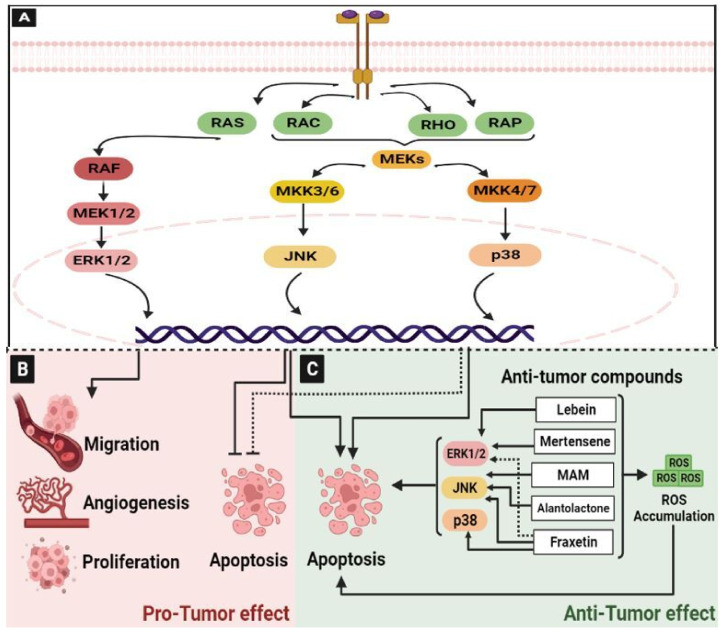
The MAPK signaling pathway in colorectal cancer. (**A**) Schematic representation of activated MAPK pathway: The interaction of mitogens and growth factors with the receptor tyrosine kinase promotes its activation and dimerization. Then, it activates the small GTPases (Ras, Rac, RHO, or RAP) subsequent to the activation of MAPKKK (Raf or MEKs) and MAPKK (MEK1/2, MKK3/6, or MKK6/7). Finally, MAPKK activates ERK1/2, p38, or JNK, and they are translocated to the nucleus to interact and activate many transcription factors, leading to the target genes’ transcription. (**B**) MAPK pathway is involved in many oncogenic processes such as cell migration, angiogenesis, and tumor proliferation. In addition, it can inhibit apoptosis. (**C**) MAPK also exhibited anti-proliferative and pro-apoptotic activities in CRC. Many natural anti-tumor compounds induce tumor cell apoptosis through ERK1/2, JNK, or/and p38. Indeed, Lebein and Mertensene activate ERK1/2, MAM and Alantolactone activate JNK, and Fraxetin induces ERK1/2, JNK and p38 phosphorylations. These compounds increase the production of ROS and induce their accumulation, thereby promoting apoptosis. Abbreviations: RAS: Rat Sarcoma, RAF: Rapidly Accelerated Fibrosarcoma, MEK and MKK: Mitogen-activated protein kinase kinase, ERK: Extracellular signal-regulated protein kinase, JNK: Jun N-terminal kinase, ROS: Reactive oxygen species, MAM: 2-methoxy-6-acetyl-7-methyljuglone.

**Figure 3 ijms-24-05600-f003:**
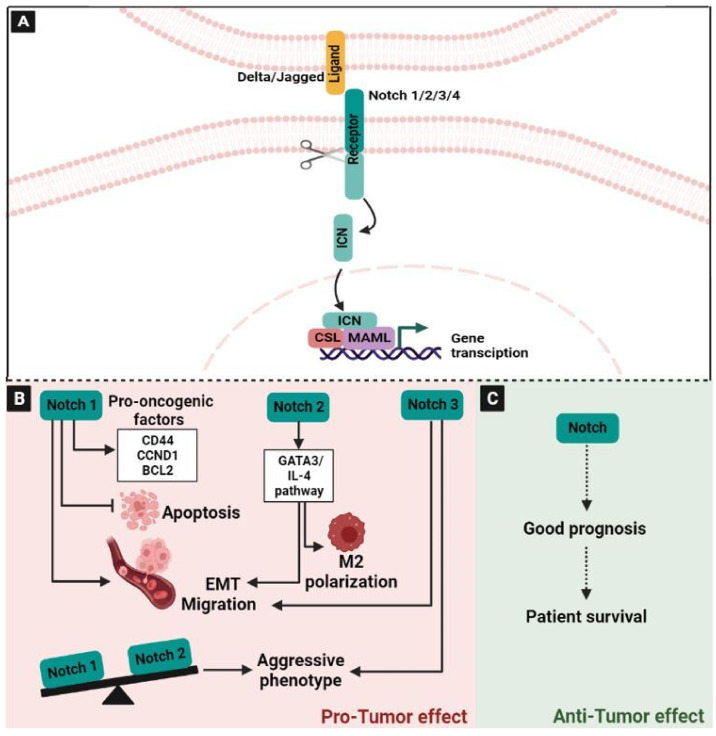
The NOTCH signaling pathway in colorectal cancer. (**A**) Schematic representation of activated NOTCH pathway: After the interaction of the ligand and the receptor, a cascade of proteolytic cleavage is started, resulting in the release of a constitutively active ICN fragment. Then, the ICN fragment is translocated to the nucleus, where it associates with CSL and MAML as part of a larger transcriptional complex. (**B**) Notch1 inhibits apoptosis and induces invasiveness and metastasis in CRC by activating several pro-oncogenic factors, such as CD44, CCND1, and the BCL2. Notch 2 activates the GATA3/IL-4 pathway, thereby promoting macrophage M2 polarization and enhancing EMT. Moreover, Notch1/2 had an opposite clinical outcome, where the increased expression of Notch1 and a reduced Notch2 were associated with poor overall survival. Notch 3 is linked to metastatic CRC and it is associated with an aggressive phenotype. (**C**) Notch2 was considered to be a good prognosis marker for CRC patients. Abbreviations: NOTCH: Neurogenic locus notch homolog, ICN: Intracytoplasmic nick, CSL: “CBF-1, Suppressor of Hairless, Lag-2”, MAML: Mastermind-like, EMT: Epithelial–Mesenchymal Transition, M2: Macrophage 2, CD: Cluster of differentiation, CCND1: Cyclin D1, BCL2: B-cell lymphoma 2, GATA3: GATA Binding Protein 3, IL: interleukin.

**Figure 4 ijms-24-05600-f004:**
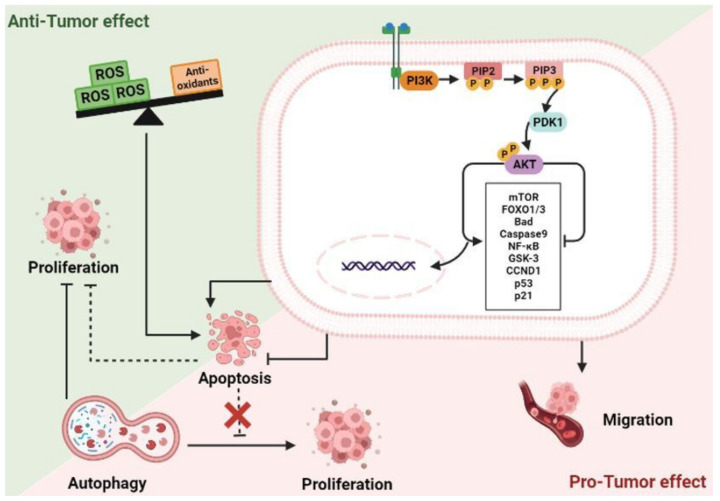
The PI3K/AKT signaling pathway in colorectal cancer. PI3K/AKT plays a double role in cancer as a survival and deadly signaling. Activated AKT regulates cell proliferation, differentiation, migration, and apoptosis in CRC by activating or inhibiting downstream target proteins, such as Bad, Caspase 9, NF-κB, GSK-3, FOXO1/3, p21, and p53. It can suppress tumor proliferation also by decreasing the expression of antioxidant enzymes and increasing the production of ROS, thereby inducing cell apoptosis. In contrast, PI3K/AKT could modulate the expression of pro-apoptotic factors to inhibit apoptosis. Autophagy is another process controlled by PI3K/AKT/mTOR signaling. It is a double-edged sword in cancer. It can inhibit or induce cell proliferation. Abbreviations: PIP2: Phosphatidylinositol 4,5-bisphosphate, PIP3: Phosphatidylinositol (3,4,5)-trisphosphate, PDK1: phosphoinositide-dependent protein kinase-1, mTOR: Mammalian target of rapamycin, Foxo: Forkhead box O, Bad: Bcl-2-associated death, NF-κB: Nuclear factor-kappaB, GSK-3: Glycogen synthase kinase 3, CCND1: Cyclin D1, ROS: Reactive oxygen species.

**Figure 5 ijms-24-05600-f005:**
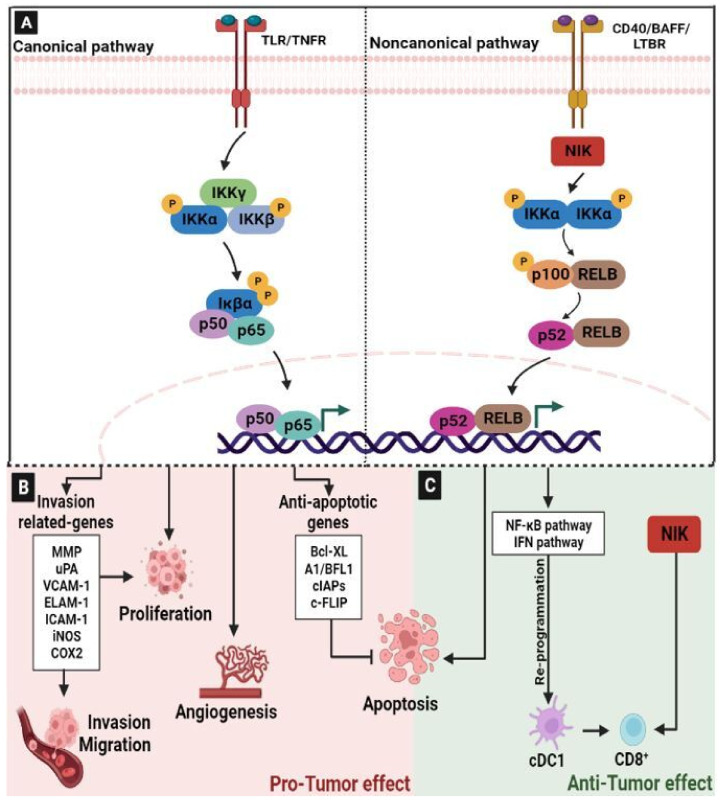
The NF-κB signaling pathway in colorectal cancer. (**A**) Schematic representation of activated NF-κB pathway: The canonical pathway is induced by TLR and TNFR. Activation of this cascade leads to the degradation of inhibitory protein IκB through its phosphorylation by a trimeric complex composed of IKKα and IKKβ, and IKKγ. The IKK complex phosphorylates p65/p50-bound IκB then translocates into the nucleus. The non-canonical pathway is dependent on the activation of the p100/ RelB complex by BAFFR, CD40, and LTBR. This cascade induces the phosphorylation of NIK, which subsequently phosphorylates IKKα. Then, p52-RelB heterodimer is activated and translocates to the nucleus. (**B**) NF-κB signaling plays a significant role in the tumorigenesis process via the regulation of downstream NF-κB gene products linked to cell growth, inflammation, metastasis, angiogenesis, and the inhibition of apoptosis. It up-regulates the expression of invasion related-genes (MMP, uPA, VCAM-1, ELAM-1, ICAM-1, iNOS, and COX2) and anti-apoptotic genes (Bcl-KL, A1/BFL1, cIAPs, and cFLIP). (**C**) NF-κB and IFN pathways also play an important role in the re-programmation of conventional dendritic cells into cDC1, leading to effective recruitment and the activation of CD8+ T cells. The non-canonical pathway of NF-κB through NIK plays an anti-tumor role by increasing glycolytic metabolism and the effector functions of CD8+ T cells. Abbreviations: TLR: Toll-like receptors, TNFR: Tumor necrosis factor receptor, IKK: IkappaB kinase, Iκβα: Nuclear factor-kappa B inhibitor alpha, CD40: Cluster of differentiation 40, BAFF: B-cell activating factor, LTBR: Lymphotoxin beta receptor, NIK: NF-κB-inducing kinase, MMP: Matrix metalloproteinasec, uPA: Urokinase-type plasminogen activator, VCAM-1: Vascular cell adhesion molecule 1, ELAM-1: Endothelial leukocyte adhesion molecule 1, ICAM-1: Intercellular adhesion molecule 1, iNOS: Inducible nitric oxide synthase, COX2: Cyclooxygenase-2, Bcl-XL: B-cell lymphoma extra-large, A1/BFL1: Bcl-2-related protein A1, cIAPs: Cellular inhibitors of apoptosis, c-FLIP: Caspase-8/FAS-associated death domain-like IL-1beta-converting enzyme inhibitory protein, IFN: interferon, cDC1: Conventional dendritic cell 1.

**Figure 6 ijms-24-05600-f006:**
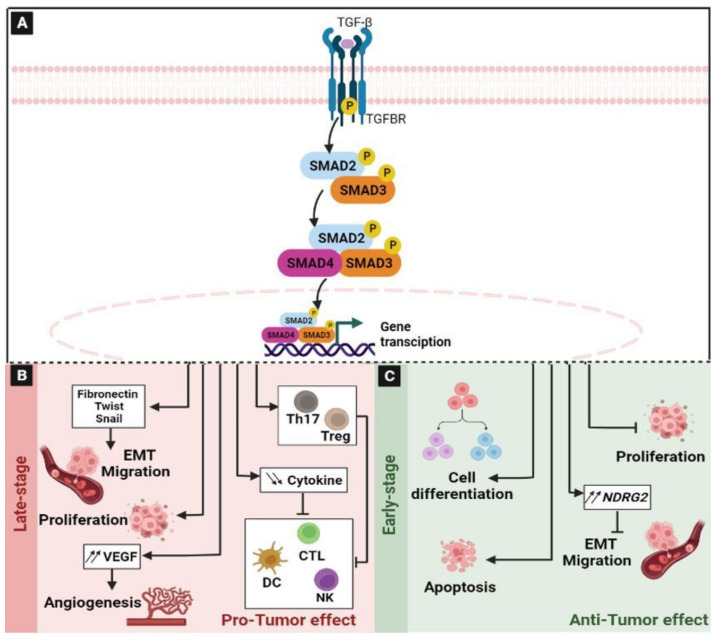
The paradoxical role of TGF-β signaling pathway in colorectal cancer. (**A**) Schematic representation of activated TGF-β pathway: The ligation of TGF-β to the receptor induces the formation of a heterotetrameric complex of TGFBRs. The activation of TGFBR1 by phosphorylation leads to phosphorylating the SMAD2/3, which can be assembled into complexes with SMAD4. Then, the complex is translocated to the nucleus where it can regulate the expression of target genes. (**B**) During the late stage, TGF-β acts as an oncogene and increases tumor proliferation. It induces the EMT, accelerating tumor invasion and metastasis through Fibronectin, Twist, and Snail. In addition, TGF-β enhances the expression of VEGF, inducing angiogenesis. TGF-β decreases cytokine production, thereby suppressing the immune system and inducing cell differentiation into Treg and Th17. (**C**) At the early stage of cancer, TGF-β acts as a tumor suppressor by inhibiting cell proliferation and EMT. It also induces apoptosis and stimulates cell differentiation. Abbreviations: TGFBR: Transforming growth factor-beta receptor, SMAD: Suppressor of Mother Against Decapentaplegic, VEGF: Vascular Endothelial Growth Factor, DC: Dendritic cell, CTL: Cytotoxic T Lymphocyte, NK: Natural Killer, NDRG2: N-Myc downstream-regulated gene 2, EMT: Epithelial–Mesenchymal Transition, Th: T helper, Treg: T regulator.

**Figure 7 ijms-24-05600-f007:**
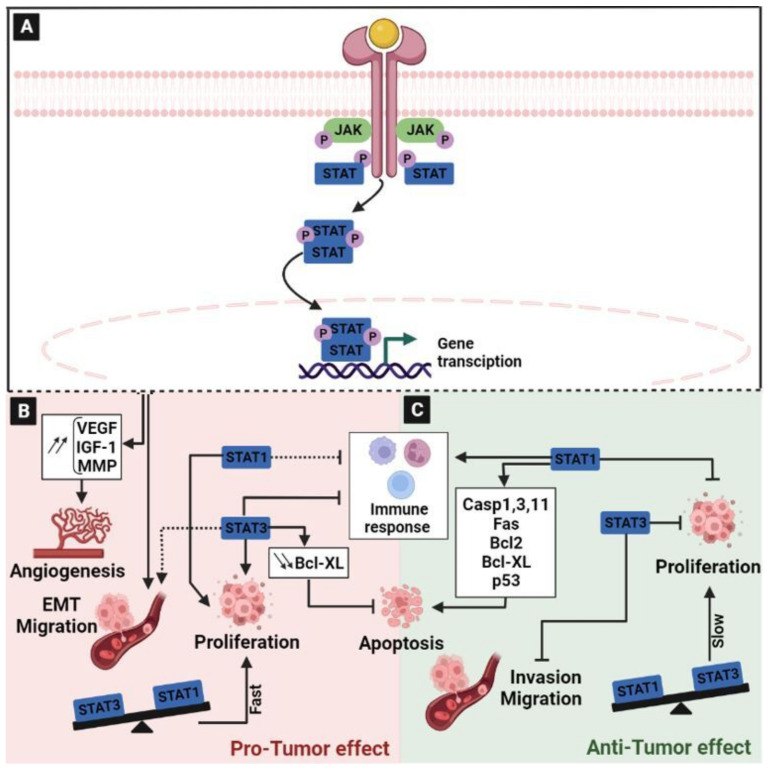
The anti-tumorigenic and the pro-tumorigenic effects of the JAK/STAT signaling pathway. (**A**) Schematic representation of activated JAK/STAT pathway: The interaction of growth factor or cytokines with the receptor induces its oligomerization, leading to the recruitment of related JAKs. Once activated, JAKs serve as docking sites for STAT. Then, STATs dissociate from the receptor to form homodimers or heterodimers, and translocate to the nucleus to initiate the transcription of a repertoire of target genes. (**B**) JAK/STAT acts as an oncogene and enhances cell migration and angiogenesis. STAT1 induces cell proliferation and inhibits the immune system. STAT3 inhibits anti-tumor response and apoptosis by down-regulating the apoptotic protein Bcl-xl. In addition, a low STAT1/high STAT3 ratio highlighted faster tumor growth. (**C**) JAK/STAT acts as a tumor suppressor through the inhibition of cell proliferation and invasion. It also induces cell death and enhances the immune response. Indeed, STAT1 induces cell apoptosis by activating the apoptotic caspases 1/3/11, p53, Fas, Bcl-2, and Bcl-X. It also inhibits tumor proliferation and the anti-tumor immune response. STAT3 exhibits an anti-tumor effect by affecting cell viability and cell migration. Furthermore, a low STAT3/high STAT1 ratio highlighted slower tumor growth. Abbreviations: VEGF: Vascular Endothelial Growth Factor, EMT: Epithelial–Mesenchymal Transition, MMP: Matrix metalloproteinase, IGF: Insulin-like growth factor, Bcl-XL: B-cell lymphoma extra-large, Casp: Caspase, Bcl2: B-cell lymphoma 2.

**Figure 8 ijms-24-05600-f008:**
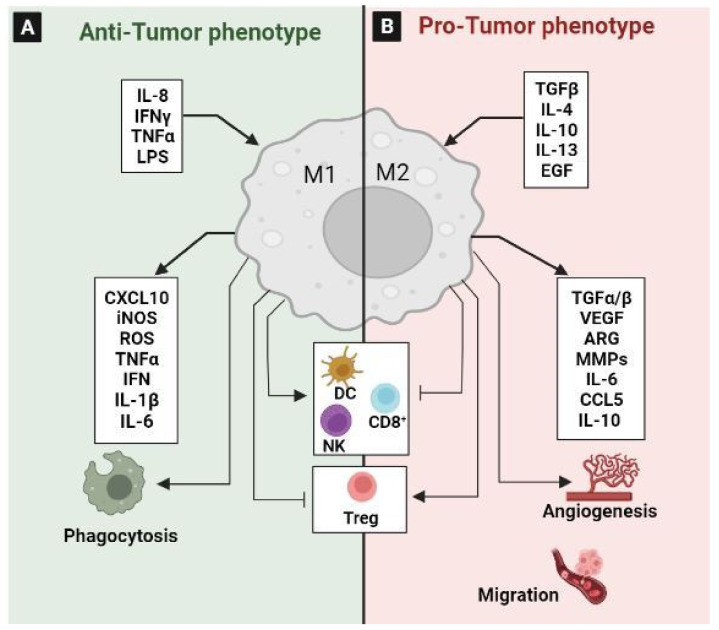
The role of tumor-associated macrophages (TAM) in the tumor microenvironment. Macrophages polarize to anti-tumoral M1 and pro-tumoral M2 phenotypes: Depending on the microenvironmental stimuli, TAMs exhibit anti-tumorigenic “M1” and the pro-tumorigenic “M2” phenotypes. (**A**) M1 macrophages are stimulated by IL-8, IFNγ, TNFα, or LPS. The stimulation of M1 macrophages triggers many downstream factors, such as CXCL10, iNOS, ROS, TNFα, IFN, IL-1β, and IL-6. These factors activate a pro-inflammatory response, stimulate tumor cell recruitment into tumors and inhibit Treg recruitment. In addition, M1 macrophages can exhibit an anti-tumor effect by phagocytosing tumor cells. (**B**) M2 macrophages are stimulated by TGFβ, IL-4, IL-10, IL-13, or EGF. This activation leads to the induction of downstream factors such asTGFα/β, VEGF, ARG, MMPs, IL-6, CCL5, and IL-10, which allow the M2 macrophage to play the role of a pro-tumor. Indeed, M2 macrophages induce angiogenesis and cell migration. They exert the function of an immunosuppressor by enhancing the Treg recruitment into the tumor. Abbreviations: TAM: Tumor associated macrophage, IL: interleukin, IFN: interferon, TNF: tumor necrosis factor, LPS: Lipopolysaccharide, TGF: Tumor growth factor, CXCL: Chemokine CXC ligand, iNOS: Inducible nitric oxide synthase, ROS: Reactive oxygen species, ARG: Arginase, VEGF: Vascular Endothelial Growth Factor, MMP: Matrix metalloproteinase, NK: Natural killer, DC: Dendritic cell, Treg: Regulatory T cell, EGF: Epidermal growth factor, CCL: CC chemokine ligand.

**Figure 9 ijms-24-05600-f009:**
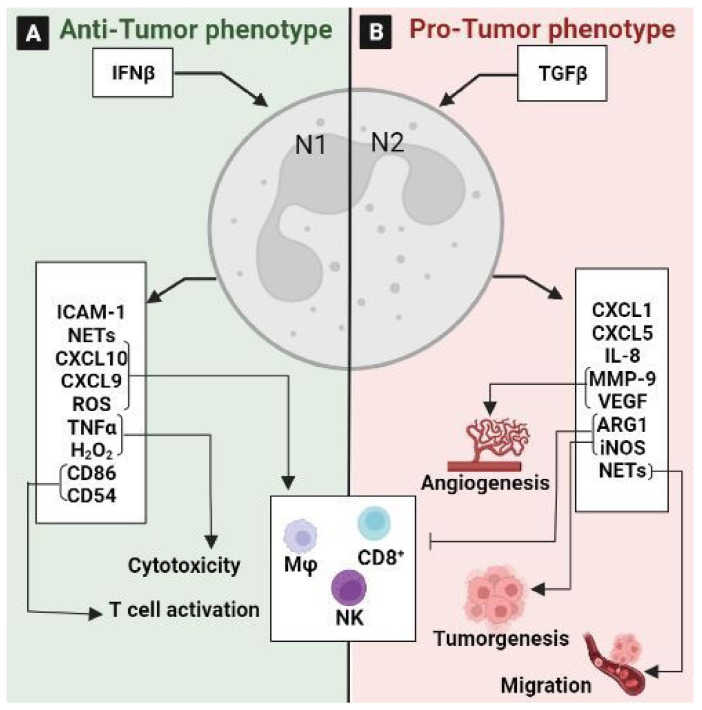
The role of tumor-associated neutrophils (TANs) in the tumor microenvironment. Neutrophils polarize to anti-tumoral N1 and pro-tumoral N2 phenotypes: Depending on microenvironmental stimuli, TAN can be polarized into two sub-populations. (**A**) Upon binding to the receptor, IFNβ stimulated N1 and activated several transcription factors involved in the anti-tumoral effect of the N1 by increasing cytotoxicity through the production of TNFα and H_2_O_2_. Many downstream factors are activated by N1 neutrophils (such as ICAMs, NETs, CXCL9/10, ROS, CD86, and CD54) to induce T-cell activation and enhance the effective immune response. (**B**) Upon the activation of N2 by TGFβ, various effectors are secreted to induce pathological angiogenesis (MMPs and VEGF) and cell migration (NETs), contributing thus to the inhibition of the anti-tumor immune response (ARG1 and iNOS). Abbreviations: TAN: Tumor-associated neutrophil, IL: interleukin, IFN: interferon, ICAM: Intercellular adhesion molecule, NETs: Neutrophil extracellular traps, CXCL: chemokine CXC ligand, ROS: reactive oxygen species, TNF: tumor necrosis factor, CD: Cluster of differentiation, Mϕ: Macrophage, NK: Natural killer, CCXL: C-X-C motif chemokine ligand, TGF: Tumor growth factor, HMGB: High mobility group box protein, MMP: matrix metalloproteinase, VEGF: Vascular endothelial growth factor, Arg1: Arginase 1, iNOS: Inducible nitric oxide synthase.

**Figure 10 ijms-24-05600-f010:**
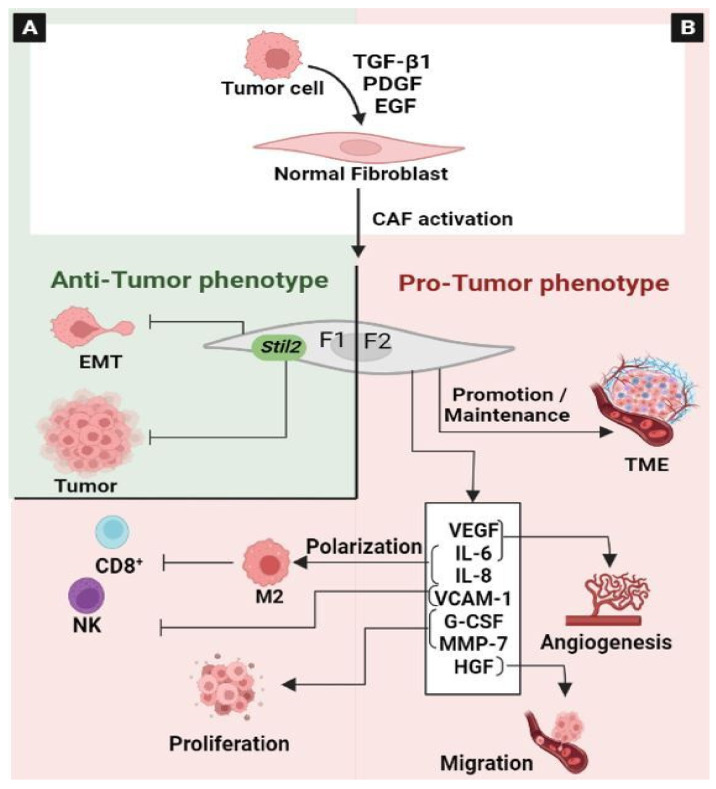
The role of cancer-associated fibroblasts (CAF) in the tumor microenvironment. Fibroblasts are stimulated by tumor cells that secrete TGF-β1, PDGF, or EGF. This activation promotes the polarization of CAF into antitumoral F1 and pro-tumoral F2. (**A**) F1 inhibits tumor proliferation and EMT through Stil2. (**B**) The polarization into F2 induces the expression of several factors, thereby exacerbating tumor growth (GSF and MMP-7), angiogenesis (VEGF and IL-6), cell migration (HGF), and promoting and maintaining the TME. In addition, it inhibits the immune response by producing VCAM-1 and inducing M2 polarization through IL-6 and IL-8. Abbreviations: IL: Interleukin, TGF: Tumor growth factor, PDGF: Platelet-derived growth factor, EGF: Epidermal growth factor, EMT: Epithelial–mesenchymal transition, TME: Tumor microenvironment, VEGF: Vascular endothelial growth factor, VCAM: Vascular cell adhesion molecule, HGF: Hepatocyte growth factor, G-CSF: Granulocyte-colony stimulating factor, MMP: matrix metalloproteinase, NK: Natural killer, M2: Macrophage 2.

**Figure 11 ijms-24-05600-f011:**
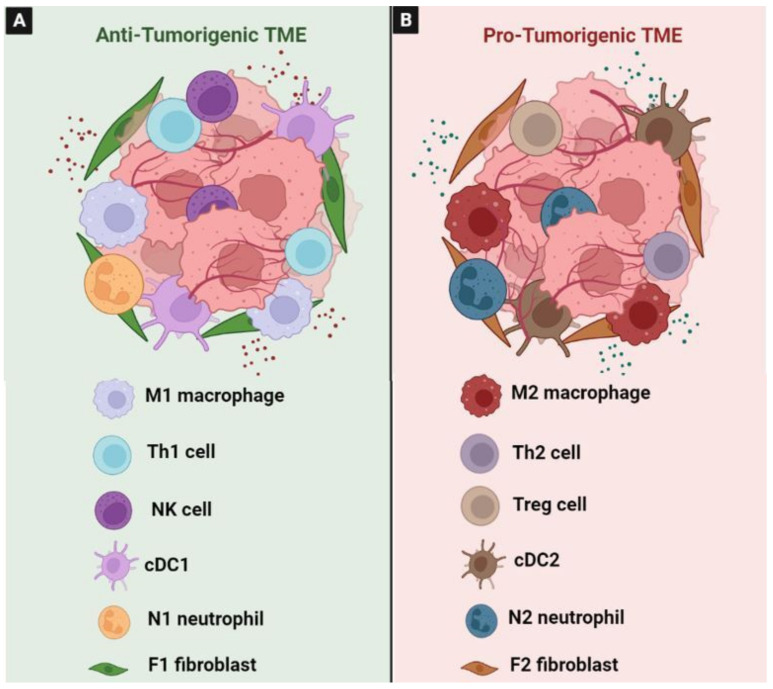
The anti-tumorigenic and the pro-tumorigenic effects of the TME components: (**A**) The anti-tumorigenic TME is characterized by the presence of cells that possess anti-tumor functions: M1 macrophages, Th1 cells, NK cells, cDC1, N1 neutrophils and F1 fibroblasts. (**B**) On the other hand, the pro-tumorigenic TME is composed mainly of cells, which have a pro-tumor effect: M2 macrophages, Th2 cells, Treg cells, cDC2, N2 neutrophils and F2 fibroblasts. Abbreviations: TME: Tumor Microenvironment, Th: T helper, NK: natural killer, cDC: Conventional Dendritic cell, Treg: T regulator.

## Data Availability

Not applicable.

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
