# Peer review of "Interplay between Signaling Pathways and Tumor Microenvironment Components: A Paradoxical Role in Colorectal Cancer"

_ijms, 2023, doi:10.3390/ijms24065600_

Round 1

Reviewer 1 Report

This review is a comprehensive and detailed description of kinase pathways, signalling factors and infiltrating cells in CRC tumours determining their microenvironment and interactions.

CRC TME is infiltrated by the heterogeneous population of immune cells playing both pro- and anti-tumour roles that depends on their polarization induced by tumours cells and modulated by complex interconnected signalling.

Authors described in detail the signalling mainly involved in CRC development and regulating the antithetical role of infiltrating cells and the immune cells populating TME.

The amount and complexity of informations implies the use of acronyms to simplify the text, although often excessive simplifications compromise the readability of the text itself and its understanding. Unfortunately, excessive exemplification connotes this review in many parts of the text, in the figures and related figures legends.

This review is filled with information related to cells and signalling pathways that regulate tumour development and TME, particularly CRC TME towards an anti-tumour or pro-tumour phenotype. The antithetical functions described and the number of data reported is such that it is hard to imagine that any result reported can be represented in the figures and explained in the figure legends. However, the figures help to understand and memorize the fundamental mechanisms of cancer development and TME, so they help acquire, for each issue, the essential meaning that can be of interest to the development of novel therapeutic protocols, which is the ultimate goal of a review like this.

The authors have done a great deal of work in writing the manuscript but they still have to make an effort to make it readable in the text and in the corresponding figures. To this purpose authors are invited to check the match between text and related figure and figure legends

As a matter of fact, one of the greatest concerns about the manuscript is that, despite the large body of information, the authors present a figure for each paragraph of signalling and tumour infiltrating cells, but in some cases they do not illustrate the mechanisms or molecules described in the text. The legends of the figures, are in these cases concise and do not clarify the all mechanism or molecules cited in the text nor help enough its understanding.

The examples of these concerns are the following:

Fig1 C . Lines 112-118 of the text intends to explain what is shown in figure 1C but it is difficult to see a correspondence between text (lines 112-118), figure and figure legend  as they are all a simplified synthesis of the pathways.

Fig 2C. The text describes the role of some compounds in the induction of colon cancer cells death. Then they described the role of JNK pathway in CRC. Do these compounds activate JNK pathway? The text nor the figure and cited bibliography explain whether JNK is activated by these compounds. Is not clear whether the all compounds cited in this paragraph, with the exception of the natural naphthoquinone MAM validated in HT29, exert their anti-apoptotic action by inducing JNK. Please clarify.

Fig 3. This figure should represent the role of Notch signalling in CRC. The text of the paragraph on Notch is mostly dedicated to other cancers and only the last part summarizes the role on CRC. The legend is also very concise. The data available are not enough but authors must attempt to ameliorate the understanding and the correlation between text and figure with the help of legend.

The reviewer has limited the observation of what is missing only in the 3 figures since the work of improving the correspondence of the text with the cited bibliography, with figures and figures legend is the task of the authors.

One more concern is the conclusion. After such detailed exposure of signalling and tumour infiltrating cells capable of playing opposite roles, one would expect more consistent conclusions and perspectives than has been done. Authors are invited to revise or better rewrite the conclusion.

Some sentences need English correction, please check.

Author Response

Responses to the reviewers´ comments Reviewer 1: Comments and Suggestions for Authors This review is a comprehensive and detailed description of kinase pathways, signalling factors and infiltrating cells in CRC tumours determining their microenvironment and interactions. CRC TME is infiltrated by the heterogeneous population of immune cells playing both pro- and anti-tumour roles that depends on their polarization induced by tumours cells and modulated by complex interconnected signalling. Authors described in detail the signalling mainly involved in CRC development and regulating the antithetical role of infiltrating cells and the immune cells populating TME. The amount and complexity of informations implies the use of acronyms to simplify the text, although often excessive simplifications compromise the readability of the text itself and its understanding. Unfortunately, excessive exemplification connotes this review in many parts of the text, in the figures and related figures legends. This review is filled with information related to cells and signalling pathways that regulate tumour development and TME, particularly CRC TME towards an anti-tumour or pro-tumour phenotype. The antithetical functions described and the number of data reported is such that it is hard to imagine that any result reported can be represented in the figures and explained in the figure legends. However, the figures help to understand and memorize the fundamental mechanisms of cancer development and TME, so they help acquire, for each issue, the essential meaning that can be of interest to the development of novel therapeutic protocols, which is the ultimate goal of a review like this. The authors have done a great deal of work in writing the manuscript but they still have to make an effort to make it readable in the text and in the corresponding figures. To this purpose authors are invited to check the match between text and related figure and figure legends As a matter of fact, one of the greatest concerns about the manuscript is that, despite the large body of information, the authors present a figure for each paragraph of signalling and tumour infiltrating cells, but in some cases they do not illustrate the mechanisms or molecules described in the text. The legends of the figures, are in these cases concise and do not clarify the all mechanism or molecules cited in the text nor help enough its understanding. We thank the reviewer for his/her constructive comments and suggestions. We appreciate a lot the interesting critics he/she raised that were helpful to improve the quality of the review. We hope that our response will satisfy the interesting questions raised. The examples of these concerns are the following: 1/ Fig1 C . Lines 112-118 of the text intends to explain what is shown in figure 1C but it is difficult to see a correspondence between text (lines 112-118), figure and figure legend as they are all a simplified synthesis of the pathways. We have revised this part according to the reviewer suggestion. Modifications are highlighted in blue in the text file. A new revised figure 1 is uploaded in page 4 and the corresponding legend is modified accordingly (Lines 137-141, 145-146). 2/ Fig 2C. The text describes the role of some compounds in the induction of colon cancer cells death. Then they described the role of JNK pathway in CRC. Do these compounds activate JNK pathway? The text nor the figure and cited bibliography explain whether JNK is activated by these compounds. Is not clear whether the all compounds cited in this paragraph, with the exception of the natural naphthoquinone MAM validated in HT29, exert their anti-apoptotic action by inducing JNK. Please clarify. As suggested by the reviewer, we revised this part by adding more explanations about how the listed natural compounds activate some signaling pathways to induce cell death. The compounds cited as examples (Lebein, Mertensene) do not activate JNK pathway. Only Fraxetin, a natural compound extracted from Fraxinus spp, induced apoptotic cell death in HT29 and HCT116 through Mitochondria Dysfunction associated to ROS induction and the modulation of ERK1/2, JNK, and P38 signaling pathways. - Lebein, an heterodimeric disintegrin isolated from Macrovipera lebetina snake venom induces the activation of ERK1/2 through reactive oxygen species (ROS) accumulation and p53-dependent apoptotic pathway in human colon adenocarcinoma cells. We recently found that this protein induced also AKT and NFkB activation (unpublished results) at least in part through ROS accumulation. - Mertensene triggered cell apoptosis through the modulation of intracellular ROS levels associated to the activation of the ERK1/2 in HT29 cells. - Modifications are highlighted in blue in the text file (Lines 176-186). - A new revised figure 2 is uploaded in page 6 and the corresponding legend is modified accordingly (Lines 210-214, 216-217). 3/ Fig 3. This figure should represent the role of Notch signalling in CRC. The text of the paragraph on Notch is mostly dedicated to other cancers and only the last part summarizes the role on CRC. The legend is also very concise. The data available are not enough but authors must attempt to ameliorate the understanding and the correlation between text and figure with the help of legend. As suggested by the reviewer, in the revised version; we modified the figures and their corresponding legends to add more clarifications and support what is written and compiled in the text file. A new revised figure 3 is uploaded in page 8 and we modified the corresponding legend accordingly (Lines 285-289, 292-293). 4/ The reviewer has limited the observation of what is missing only in the 3 figures since the work of improving the correspondence of the text with the cited bibliography, with figures and figures legend is the task of the authors. All the reviewer comments and suggestions were taken into account. Figures and their corresponding legends were modified accordingly in the revised manuscript. - A new revised figure 1 is uploaded in page 4 and the corresponding legend is modified accordingly (Lines 137-141, 145-146). - A new revised figure 2 is uploaded in page 6 and the corresponding legend is modified accordingly (Lines 210-214, 216-217). - A new revised figure 3 is uploaded in page 8 and we modified the corresponding legend accordingly (Lines 285-289, 292-293). - A new figure 4 is uploaded in page 9 and the corresponding legend is modified accordingly (Lines 340-342, 349-350). - A new figure 5 is uploaded in page 11 and the corresponding legend is modified accordingly (Lines 422-427, 431-436). - A new figure 6 is uploaded in page 13 and the corresponding legend is modified accordingly (Lines 501-505, 510). - A new figure 7 is uploaded in page 16 and the corresponding legend is modified accordingly (Lines 610-616, 618-619). - A new figure 8 is uploaded in page 18 and the corresponding legend is modified accordingly (Lines 682-688, 694). - A new figure 9 is uploaded in page 19 and the corresponding legend is modified accordingly (Lines 732-738). - A new figure 10 is uploaded in page 21 and the corresponding legend is modified accordingly (Lines 805-810, 814). All Modifications are highlighted in blue in the text file 5/ One more concern is the conclusion. After such detailed exposure of signalling and tumour infiltrating cells capable of playing opposite roles, one would expect more consistent conclusions and perspectives than has been done. Authors are invited to revise or better rewrite the conclusion. We agree with the reviewer concerning the need of rewriting the conclusion to support the reported data. We have revised the conclusion section accordingly (page 25. lines 955-970). Modification is highlighted in blue in the text file 6/ Some sentences need English correction, please check. Corrections were made in the text file. Modifications are highlighted in blue.

Reviewer 2 Report

The authors of the ijms-2258545 manuscript set themselves the difficult task of writing a review article on the cross talk between colorectal cancer (CRC) signaling system and its microenvironment (TME).  Such communication between tumor and TME is establish through paracrine factors that induce and propagate activation of key signaling pathways. Colorectal cancer, its carcinogenesis and maintaining malignant phenotype especially and critically depends on its TME.

This is a very conscientiously and reliably but also carefully written work. The authors show and emphasize the double function of such interactions, which can be both pro- and anti-tumor, sort of double-edged sword function.  

They emphasize its complexity and the unpredictability. They also managed to update and compile the latest published results, which further prove the importance of the interaction between CRC and TME for the development of the disease.

This is a very long manuscript of 34 pages. Analyzing its rich content, I did not find any substantive or linguistic shortcomings. In my opinion, this manuscript is suitable for publication in IJMS.

Author Response

We thank a lot the reviewer for his/her appreciation and for the positive feedback and comment.

Reviewer 3 Report

The Authors of the manuscript Sonia Ben Hamouda and Khadija Essafi-Benkhadir have done a commendable job in their manuscript entitled" Interplay between signaling pathways and tumor microenvironment components: a paradoxical role in colorectal cancer". The authors have written a comprehensive account of some of the signaling pathways in CRC development and progression. Additionally the authors have also emphasized on the cross talk between tumor cells and the TME. This account overall helps the readers to get a wholesome background as well as helps in documenting the roles orchestrated by these processes. This review also serves s a review to refining  strategies for medical intervention with respect to CRC and in revisiting the treatment plans currently offered. The figures also enhance the readers engagement with the many signaling pathways highlighted and their role in colorectal cancer.

Author Response

(The authors gave the same response as above.)
